# IPPRO: Importance-based Pruning with PRojective Offset for Magnitude-indifferent Structural Pruning

## Abstract

Not only the classical methods of neural network pruning but also most importance-based pruning methods rely too much on parameter magnitudes to prune effectively. We propose a novel pruning strategy, named IPPRO, using projective space to alleviate the unfair advantage given to parameter magnitudes. We use gradient of loss in the projective space to construct PROscore, which is a magnitude-indifferent score that is in turn used by IPPRO, our novel importance-based structured pruning algorithm. Extensive experiments on Convolutional Neural Networks (CNNs), Vision Transformers (ViT), and Large Language Models (LLMs) demonstrate that IPPRO consistently outperforms, especially in high compression scenarios. Our results establish IPPRO as a task-agnostic and architecture-agnostic pruning paradigm, offering both a new theoretical foundation and a practical tool for magnitude-indifferent structural pruning.

## 1 Introduction

Deep neural networks have achieved remarkable success across vision and language models, but their rapidly increasing scale poses severe challenges for computation and deployment on resource-constrained platforms. Structured pruning has emerged as a practical solution, as channel and filter-level sparsity translates directly into reduced FLOPs and memory.

Magnitude based pruning (Li et al., 2017), which assumes that "larger filters are more important", remains dominant (Fang et al., 2024; 2023) due to its simplicity and efficiency. However, this heuristic introduces fundamental limitation: as shown the dot-and-dash line in Fig. 1, pruning decisions are strongly biased by filter norms, leading to sensitivity under different normalization schemes and significant performance degradation at high compression ratios. More advanced criteria have been proposed, such as gradient information via Taylor expansion (Molchanov et al., 2019; Ma et al., 2023) or geometric similarity of filter clusters (He et al., 2019). However, as shown in Fig. 2, these methods are still not fully independent of magnitude information, as indicated by the gap between the red and green verticals. Except for IPPRO, no prior pruning method

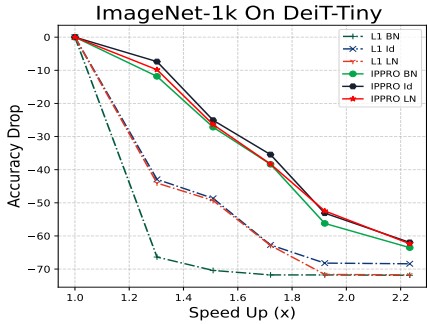

Figure 1: Pruning performance comparison without fine-tuning of IPPRO and Magnitude pruning on the pre-trained DeiT-Tiny model with different normalizations (BatchNorm, IdentityNorm, LayerNorm).

has provided a fundamental solution that completely departs from "size matters" criteria.

We address this issue by observing the dynamics of filters through projective geometry, which considers the space consist of equivalence classes under scale invariance. Within lens of projective geometry, we can discard the role of magnitude from the filter and consider the magnitude-invariant criteria which defined via dynamics in projective space. In practice, we consider the projective offset from the origin as similarity, placing filters on equidistance location and observe the filters move

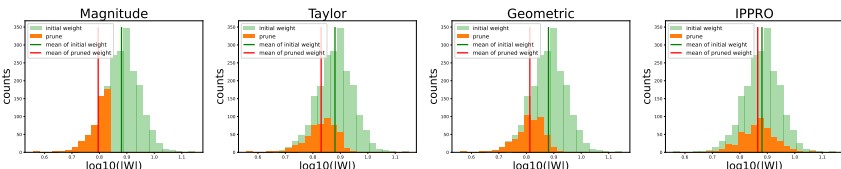

Figure 2: Visualization of magnitude of pruning filters obtained by four different criteria, on DeepLabV3-Resnet50 with Cityscapes dataset.

toward origin or not. We select this post-movement offset from origin as filter importance criteria and propose IPPRO: the Importance-based Pruning with PRojective Offset.

IPPRO is conceptually simple, straightforward to implement, and model architecture agnostic. We validate its practical effectiveness across diverse architectures, ranging from Convolutional Neural Networks (CNNs) and Vision Transformers (ViT) based vision models to Large Language Models (LLMs), where it consistently achieves strong performance even under high-pruning regimes and in no fine-tuning scenarios. We formally define the theoretical foundations of PROscore, the key importance score with magnitude-indifference in Section 3, describe implementation of IPPRO in Section 4, and present its comprehensive empirical evaluation in Section 5.

## 2 RELATED WORKS

### 2.1 STRUCTURED PRUNING USING MAGNITUDES

Deep Neural Networks (DNNs) are often overparameterized, leading to unnecessary computational costs. Pruning addresses this by removing unimportant weights or structures. Early methods like magnitude-based pruning (Han et al., 2015) eliminate weights with small absolute values, achieving model compression with minimal accuracy loss. However, such approaches may overlook a weight's actual impact on performance.

To improve pruning effectiveness, gradient-based methods (Blalock et al., 2020; Molchanov et al., 2019) consider how changes in weights affect the loss function, enabling more informed decisions. A key limitation is that most methods rely on magnitude information, making it difficult to fully disregard its influence. So, our proposed method, IPPRO, address this limitation by moving beyond magnitude-based heuristics to offer a more nuanced, magnitude-indifferent approach to structured pruning.

### 2.2 IMPORTANCE-BASED PRUNING

Importance-based pruning evaluates filter contribution to remove redundancy in CNNs. Norm-based methods like $L_1$-norm (Li et al., 2017) and $L_2$-norm (He et al., 2018) prune low-magnitude filters, while others use filter similarity. For example, (He et al., 2019) uses the Geometric Median to remove closely clustered filters, and (Yvinec et al., 2021; 2022) apply Scalar Hashing and Input-wise Splitting to detect and prune similar filters based on input relevance.

Statistical and structural approaches further enhance pruning. (Wang et al., 2019b) uses Pearson Correlation to remove highly similar filters and applies Layer-wise Max-Normalization for cross-layer comparison. (Wang et al., 2021) treats filters as graph nodes to detect redundancy via structural properties. Recently, (Gupta et al., 2024a) introduced a torque-inspired method that weights filters by distance from a pivot, capturing spatial structure and offering a simple yet effective pruning strategy.

Our work distinguishes itself from either magnitude or statistics based approaches, as IPPRO formulates pruning in a projective geometry space with a novel importance score defined from gradient flow.

### 2.3 STRUCTURED PRUNING FOR TRANSFORMER ARCHITECTURES

Transformer architectures have become dominant in both vision and language domains, driven by the success of Vision Transformers (ViTs) and large language models (LLMs). However, pruning

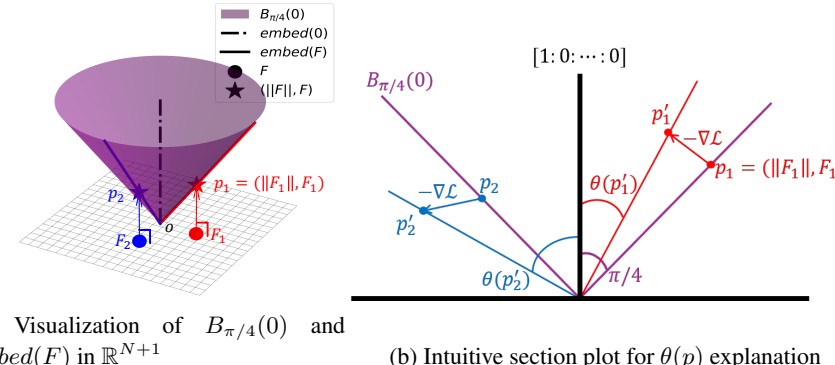

(a) Visualization of $B_{\pi/4}(0)$ and $embed(F)$ in $\mathbb{R}^{N+1}$

(b) Intuitive section plot for $\theta(p)$ explanation

Figure 3: Illustrations for conceptual understanding of PROscore for $N$-dimensional filter $F$.

transformers presents unique challenges due to the structure of multi-head self-attention (MHA). Unlike convolutional layers, where filters can be pruned independently, attention mechanisms often require coordinated pruning of related components.

To address this, two primary pruning targets have emerged in the literature. The first is *head pruning*, which removes entire attention heads to reduce computational cost and redundancy (Michel et al., 2019; Yu et al., 2022a; Yu & Xiang, 2023). The second is *neuron-level pruning*, which reduces the dimensionality within each head by removing individual neurons or projection components (Shim et al., 2024; Yu et al., 2022b; Zhu et al., 2021). Some recent approaches combine both strategies and extend pruning to additional components such as embedding dimensions (Yang et al., 2023a; Fang et al., 2024) or entire attention blocks (Yu et al., 2022b).

Although IPPRO can be applied to both head-level and neuron-level pruning by grouping related neurons, this work focuses on neuron-level pruning. This choice aligns naturally with the generalization of filter pruning in CNNs, where individual channels are evaluated and removed based on importance. Neuron-level pruning offers finer granularity and greater flexibility, making it a more suitable target for magnitude-invariant importance scoring such as PROscore.

## 3 METHODS

Let $C$ be number of prunable channels in a neural network model, and $F_1, \cdots, F_C \in \mathbb{R}^N$ be the filters which correspond to the channels. Each filter $F_i$ is a vector, which has two distinct types of geometric information: magnitude and direction. We aim to design a novel pruning methodology that challenges the "size matters" myth in pruning, where 'size' is the magnitude of a prunable filter, as this myth limits the potential of magnitude-based pruning.

In Section 3.1, we introduce the concept of projective geometry, and define the mapping function $embed$ which maps the filters to the projective space $\mathbb{RP}^N$. After placing the filters to $\mathbb{RP}^N$, in Section 3.2 we explain the distance in projective space given by an angular distance, and define the proposed importance using the angular distance.

### 3.1 EMBEDDING FILTERS INTO PROJECTIVE SPACE

To extract the directional information of a filter, which can complement its magnitude, we place the filter on a projective space with proper mapping $embed$, and specify the projective space which we utilized. In algebraic geometry, the (real) projective space $\mathbb{RP}^N$ is defined as quotient of $\mathbb{R}^{N+1} - \{0\}$ under equivalence relation $x \sim cx$ for all nonnegative real constant $c$, comprised of lines in $\mathbb{R}^{N+1}$ passing through origin. The element of $\mathbb{RP}^N$ can be expressed as $[v] = [v_0 : \cdots : v_N]$ which called homogeneous coordinates, where

$$\forall c \neq 0, [v_0 : \cdots : v_N] = [cv_0 : \cdots : cv_N]. \tag{1}$$

The point $[v]$ of projective space $\mathbb{PR}^N$ exactly corresponds to the line in $\mathbb{R}^{N+1}$ whose direction vector is parallel to $v$ and passing through 0.

Considering the filter as a vector in $\mathbb{R}^N$, we can find the embedding function which maps filter $F_i = (F_{i1}, \cdots, F_{iN}) \neq 0$ to $\mathbb{RP}^N$ by following function:

$$embed : (F_{i1}, \cdots, F_{iN}) \mapsto [\|F_i\| : F_{i1} : \cdots : F_{iN}]. \tag{2}$$

In Fig. 3a, we visualize how the filter $F$ is mapped to the projective representation in $N = 2$ case. Note that $N = 2$ appears in practice when the target layer is batch normalization layer. For the zero filter $0_F$, the embedding would be $[1 : 0 : \cdots : 0]$, which corresponds to axis line of extra parameter in $\mathbb{R}^{N+1}$. Another benefit of this placement to $\mathbb{RP}^N$ is that it naturally enjoys the magnitude-indifferent characteristic readily noticed by the scale invariance: $embed(cF) = embed(F)$ for $c > 0$.

## 3.2 PROSCORE: A NOVEL IMPORTANCE SCORE FOR PRUNING

In projective space $\mathbb{RP}^N$, the distance between two points may be considered as an angle between two lines in $\mathbb{R}^{N+1}$ where each line represents each point in $\mathbb{RP}^N$. If we denote $B_r(0)$ be the set of points in $\mathbb{RP}^N$ whose angular distance between the origin is equal to $r$, then we can see that $Im(embed) \subseteq B_{\pi/4}(0)$; i.e. the distance between $embed(F)$ and the origin is always equal to $\pi/4$. This is visualized in Fig. 3a using line representations in $\mathbb{R}^{N+1}$ that $B_{\pi/4}(0)$ becomes the cone centered at 0 and $embed(F)$ will become a line included in $B_{\pi/4}(0)$.

This phenomenon is exploited to ensure fair chance for the filters to be pruned, overcoming the sole dependency on the magnitude. After placing the filters to $B_{\pi/4}(0)$, we gauge the fate of each filters by estimating whether it would move closer to origin (and be pruned) or not, according to the direction of gradient descent. As depicted in Fig. 3b, if the gradient descent moves $p_i = embed(F_i)$ to one-step forwarded point $p_i'$ in $\mathbb{RP}^N$, then the movement toward origin can be represented by angular distance $\theta(p_i')$.

We define the tangent $\tan(\theta(p_i'))$ as a importance score for pruning decision of filter $F_i$, and name it **PRojective Offset score (PROscore)**. Precisely, given $F_i$ let $D_i$ be extra variable initialized by $\|F_i\|$. Then the projective point $p_i = [\|F_i\| : F_i]$ would be updated to Eq. (3) by gradient descent on loss function $\mathcal{L}$, and $\lambda$ which is a hyperparameter that serves a similar role to learning rate in the loss function.

$$p_i' = \left[ \|F_i\| - \lambda \frac{\partial \mathcal{L}}{\partial D_i} : F_i - \lambda \nabla_{F_i} \mathcal{L} \right] \tag{3}$$

And the PROscore is computed by

$$PROscore_\lambda(i) := tan(\theta(p_i')) = \frac{\|F_i - \lambda \nabla_{F_i} \mathcal{L}\|}{|D_i - \lambda \frac{\partial \mathcal{L}}{\partial D_i}|}, \tag{4}$$

which would be the proposed importance score of $i$-th filter. As depicted in Fig. 3b, the $i$-th channel would have small $\theta(p_i)$ when the red-colored movement of gradient descent makes $p_i$ closer and otherwise (the blue-colored arrow) not.

## 4 IMPLEMENTATION OF IPPRO

We present **Importance-based Pruning with PRojective Offset (IPPRO)**, our implementation of structural pruning using PROscore presented in Section 3. First, we overview the parameter injection trick to realize the projective space embedding. Then, we present the algorithm to compute PROscore, our novel importance score for IPPRO, and conclude the section with the remaining implementation details.

### 4.1 PROJECTIVE OFFSETTING VIA PARAMETER INJECTION

We implement the idea of embedding into the projective space as outlined in Section 3 by introducing additional parameter $D$ with desired initialization $D^{init} = diag(\|F_1\|, \cdots, \|F_N\|)$. To bring filters to projective space, we inject $D$ parameter to the model using the extension method (Jung & Lee, 2025), which adds two identical parameters $D$ and $\overline{D}$, by modifying element-wise computation layer $\sigma$, which appear next to the target layer $W$ as follows:

$$\psi_{D,\overline{D}}(x) = Dx - \overline{D}x + \sigma(x). \tag{5}$$

The modification by $\psi_{D,\overline{D}}$ does not harm the model's functionality and performance, since $D$ and $\overline{D}$ are set to be same. This parameter injection trick does not alter the model's forward functionality or final outputs, but rather serves to extract auxiliary gradient information during the backward pass, which is exclusively used for computing PROscore. The introduced parameters are removed to retrieve the original model before actual pruning happens, as they are needed for PROscore computation.

## 4.2 Computing PROscore of Filter

In Algorithm 1 we present the algorithm that we implemented to compute PROscore.

---

**Algorithm 1:** PROscore Computation

1: **Input** number of channels $N$, dataset $\mathcal{D}$, layer $W$, consecutive operation $\sigma$ and model $f(W, \sigma)$
2: Compute $D^{init} = diag(\|F_1\|, \cdots, \|F_N\|)$
3: Extend model to $f(W, \psi_{D,\overline{D}})$ using $\psi_{D,\overline{D}}$ in Eq. (5)
4: Initialize $D, \overline{D} \leftarrow D^{init}$
5: Compute $\nabla\mathcal{L}$ using backpropagation with $\mathcal{D}$
6: Compute $\tan(\theta(p'_i))$ for all $i$'s by Eq. (4)
7: **Return** the $PROscore_\lambda(i) \tan(\theta(p'_i))$ for each filter $F_i$

---

Given pretrained model with filters $F_i = W_{i,:}$, we compute $D^{init}$ by line 2 of Algorithm 1, and extend the model by adding parameters to $\sigma$, which are initialized by $D^{init}$. The choice of $\sigma$ would be followed by the choice of target layer. For example, if the target is scaling factor of batch normalization layer in ResNet then $\sigma$ would be the ReLU activation. After extending model, we estimate the loss gradient using the training dataset via backpropagation. Finally, we compute the tangent value of angular distance $\theta(p'_i)$ between the origin and $p'_i$.

## 4.3 Computing PROscore of Attention

Attention layer of transformer, mostly the MHA (Multi-Head self attention) requires several modifications to prune in sense NP (neuron pruning) which introduced in (Shim et al., 2024).

First, the MHA does not include any activation layer $\sigma$ inside the attention score computation, thus we consider the auxiliary identity activation right after the QKV computation and extend the model by replacing this identity function $\psi_D(x) = Dx + x$. In initialization of D, we consider the multiplier $m_i$ which is solution of quadratic equation $\frac{1}{m_i} - 1 = m_i \|F_i\|$ for each filter index $i$. After, we rescale $F_i$ by $F_i \leftarrow m_i F_i$ and initialize $D_i \leftarrow \frac{1}{m_i} - 1$. As a result, this initialization does not change the Q,K and V tensor but place $(D_i, F_i)$ on $B_{\pi/4}(0)$.

Second, due to latency issue, the number of neurons of each head must be equal for the parallel operation in GPU. Therefore, we group the neurons to be pruned together, and take average PROscore over each neurons.

Lastly, the number of channels in query and key must equal; we adjust the pruning ratio to be equal on query and key, when we prune the attention layer after PROscore computation.

## 4.4 Implementation Overview

IPPRO computes the PROscore without updating model parameters. Gradients are accumulated across the dataset, and pruning is applied to the original model using the precomputed scores. This design avoids iterative retraining during scoring and improves scalability. Further details of the implementation are provided in Section A. To ensure fair comparisons, we followed a standardized fine-tuning protocol, full details of which are described in Section B. In particular, we strictly adhered to the original pretrained fine-tuning recipe and did not compare against models trained with modified or alternative recipes (e.g., modified distillation loss), so that our evaluation remains consistent and unbiased.

Table 1: DeiT and EfficientFormer pruning performance on ImageNet-1k

| Model | Method | Acc(↑%) | | | Params↓(%) | FLOPs↓(%) |
|---|---|---|---|---|---|---|
| | | Base. | Prun. | Δ | | |
| DeiT-Base | WDPruning (Yu et al., 2022a) | 81.80 | 80.76 | -1.04 | 36.1 | 43.7 |
| | X-Pruner (Yu & Xiang, 2023) | 81.80 | 81.02 | -0.78 | - | 51.7 |
| | UVC (Yu et al., 2022b) | 81.80 | 80.57 | -1.23 | - | 54.5 |
| | SNP (Shim et al., 2024) | 81.80 | 79.63 | -2.17 | 63.5 | 63.6 |
| | **IPPRO (ours)** | 81.80 | **81.14** | **-0.66** | **63.7** | 63.6 |
| DeiT-Small | WDPruning (Yu et al., 2022a) | 79.85 | 78.38 | -1.47 | 39.8 | 43.7 |
| | X-Pruner (Yu & Xiang, 2023) | 79.85 | 78.93 | -0.92 | - | 47.8 |
| | UVC (Yu et al., 2022b) | 79.85 | 78.82 | -1.03 | - | 50.0 |
| | SNP (Shim et al., 2024) | 79.85 | 78.52 | -1.33 | 54.7 | **56.5** |
| | **IPPRO (ours)** | 79.85 | **79.13** | **-0.72** | **55.4** | **56.5** |
| DeiT-Tiny | SSViTE (Chen et al., 2021) | 72.20 | 70.12 | -2.08 | 26.3 | 30.7 |
| | WDPruning (Yu et al., 2022a) | 72.20 | 70.34 | -1.86 | 38.5 | 46.1 |
| | X-Pruner (Yu & Xiang, 2023) | 72.20 | 71.10 | -1.10 | - | **53.8** |
| | UVC (Yu et al., 2022b) | 72.20 | 71.30 | -0.90 | - | **53.8** |
| | SNP (Shim et al., 2024) | 72.20 | 70.29 | -1.91 | **47.3** | **53.8** |
| | **IPPRO (ours)** | 72.20 | **71.67** | **-0.53** | 43.3 | **53.8** |
| EfficientFormer-L1 | SNP (Shim et al., 2024) | 79.20 | 75.53 | -3.67 | - | **53.8** |
| | **IPPRO (ours)** | 79.20 | **77.08** | **-2.22** | 48.8 | **53.8** |

# 5 EMPIRICAL VALIDATIONS

We validate the effectiveness of our proposed method, IPPRO, across a diverse range of deep learning tasks, including image classification, semantic segmentation, and language. For image classification, we conduct experiments on DeiT (Touvron et al., 2021) and EfficientFormer (Li et al., 2022) with the ImageNet-1k dataset (Deng et al., 2009), and on various CNN models using ImageNet-1k, CIFAR-10 and CIFAR-100 (Krizhevsky et al., 2009). For semantic segmentation, we use DeepLabv3-ResNet50 (Chen et al., 2017) on the Cityscapes dataset (Cordts et al., 2016). Furthermore, to demonstrate its versatility, we apply IPPRO to a language task using the LLAMA-7B model (Touvron et al., 2023). These comprehensive experiments showcase IPPRO's robust performance and broad applicability.

IPPRO's performance was compared against various state-of-the-art pruning methods, Such as Taylor (Molchanov et al., 2019), SNP (Shim et al., 2024), SIRFP (Lv et al., 2024), and LLM-Pruner (Ma et al., 2023). Across all benchmarks, IPPRO consistently delivered better or comparable performance with a reduced model size.

## 5.1 PRUNING PERFORMANCE ANALYSIS

### 5.1.1 PERFORMANCE ANALYSIS OF IPPRO ON CNNS

We first evaluate IPPRO on CNN-based semantic segmentation task using DeepLabv3 with ResNet-50 on the CityScapes dataset (Cordts et al., 2016), following the setup of DCFP (Wang et al., 2023). While SIRFP (Wu et al., 2025) and DCFP (Wang et al., 2023) are segmentation task-specific pruning method, IPPRO is task and architecture-agnostic. Despite this generality, IPPRO achieves comparable or superior results, even showing an edge over SIRFP in some settings, which highlights the strength of our approach. As shown in Table 2, IPPRO preserves full mIoU at around 60% model reduction and consistently outperforms or matches other methods across pruning levels. Qualitative results (Figs. 4 and 7) show that semantic structure and object boundaries remain intact, even under high sparsity.

For image classification, we further validate IPPRO on ResNet-50 and MobileNet with

Table 2: Cityscapes - DeepLabV3-ResNet50

| Method | mIoU | | | Params↓(%) | FLOPs↓(%) |
|---|---|---|---|---|---|
| | Base. | Prun. | Δ | | |
| Random | 81.6 | 78.7 | -2.9 | 59.8 | 60.6 |
| NS (Liu et al., 2017) | 81.6 | 79.9 | -1.7 | 62.3 | 59.4 |
| Taylor(Molchanov et al., 2019) | 81.6 | 80.3 | -1.3 | 63.7 | 60.1 |
| DepGraph(Fang et al., 2023) | 81.6 | 80.0 | -1.6 | 59.2 | 60.4 |
| FPGM(He et al., 2019) | 81.6 | 80.2 | -1.4 | 63.9 | 61.2 |
| DCFP(Wang et al., 2023) | 81.6 | 80.9 | -0.7 | 64.2 | 60.9 |
| FGP(Lv et al., 2024) | 79.3 | 79.0 | -0.3 | 64.4 | 60.4 |
| SIRFP(Wu et al., 2025) | 81.6 | 81.3 | -0.3 | 64.8 | 61.3 |
| **IPPRO (ours)** | 81.5 | **81.5** | **0.0** | 64.0 | 61.8 |
| FPGM(He et al., 2019) | 81.6 | 79.3 | -2.3 | **74.5** | 71.0 |
| DCFP(Wang et al., 2023) | 81.6 | 80.2 | -1.4 | 74.3 | 71.3 |
| SIRFP(Wu et al., 2025) | 81.6 | 80.9 | -0.7 | 74.4 | 71.9 |
| **IPPRO (ours)** | 81.5 | **81.2** | **-0.3** | **74.5** | 72.2 |
| FPGM(He et al., 2019) | 81.6 | 77.9 | -3.7 | 84.8 | 80.7 |
| DCFP(Wang et al., 2023) | 81.6 | 79.5 | -2.1 | 83.9 | 80.2 |
| SIRFP(Wu et al., 2025) | 81.6 | 79.4 | -2.2 | **85.7** | 81.6 |
| **IPPRO (ours)** | 81.5 | **79.6** | **-1.9** | 85.1 | 81.6 |

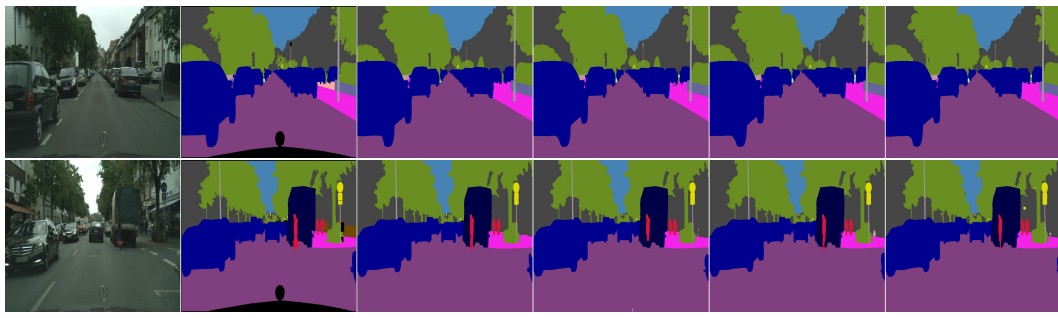

Figure 4: Qualitative results of IPPRO on the segmentation task: Cityscapes dataset using DeepLabv3. From left to right: input image, ground truth, unpruned model, and IPPRO-pruned models with 60%, 70%, and 80% sparsity, respectively.

Table 3: LLAMA-7b pruning performance comparison of various pruning method on five common reasoning datasets

| Remain Param Ratio | Method | Acc(↑%) | | | | | Avg(↑%) | Drop(↓%) |
|---|---|---|---|---|---|---|---|---|
| | | BoolQ | PIQA | ARC-e | ARC-c | OBQA | | |
| 1.0 | Baseline | 0.73 | 0.78 | 0.67 | 0.41 | 0.42 | 0.600 | 0.0 |
| 0.8 | LLM-Pruner (Ma et al., 2023) | 0.69 | 0.76 | 0.63 | 0.37 | 0.40 | 0.570 | 3.0 |
| | FLAP (An et al., 2024) | 0.69 | 0.76 | **0.69** | **0.39** | 0.39 | 0.584 | 1.6 |
| | Dobi-SVD (Wang et al., 2025) | **0.73** | **0.77** | 0.65 | 0.37 | **0.42** | 0.588 | 1.2 |
| | **IPPRO (ours)** | **0.73** | **0.77** | 0.67 | 0.38 | 0.40 | **0.590** | **1.0** |

ImageNet-1K, as well as VGG19 and ResNet-56 with CIFAR-10/100. Across all settings, IPPRO demonstrates robust compression–accuracy trade-offs. More detailed results and analyses are provided in Section C.

## 5.2 PERFORMANCE ANALYSIS OF IPPRO ON VISION TRANSFORMERS(ViT)

To demonstrate the generalizability of IPPRO beyond CNNs, we conducted a comprehensive performance comparison of recent structural pruning methods, including IPPRO, on a range of Transformer-based vision models: DeiT (Base/Small/Tiny), and EfficientFormer-L1. The experiments were performed on the ImageNet-1k dataset, and the results are summarized in Table 1.

Table 1 clearly illustrates that IPPRO demonstrates superior performance across the Various DeiT models. Notably, when compared to other pruning methods achieving similar FLOPs reductions, IPPRO consistently maintained the lowest or a highly competitive accuracy drop. For instance, on the DeiT-Base model, IPPRO reduced FLOPs by 63.6% with an accuracy drop of only 0.66%. This is a significantly better result than SNP, which showed a much larger accuracy drop of 2.17% for the same FLOPs reduction. Furthermore, IPPRO exhibited the minimal accuracy degradation among all compared methods for this model.

Our experiments on the EfficientFormer-L1 model also confirmed the superior performance of IPPRO. While SNP achieved a 53.8% FLOPs reduction with a substantial accuracy drop of 3.67%, IPPRO maintained more stable performance with a smaller accuracy drop of 2.22%. This result highlights the ability of the IPPRO methodology to generalize effectively to other Transformer-based models for vision tasks, showcasing its broad applicability.

## 5.3 PERFORMANCE ANALYSIS OF IPPRO ON LARGE LANGUAGE MODELS(LLMS)

To demonstrate that IPPRO is not limited to vision tasks and can generalize to a variety of other tasks, we conducted experiments on both the LLaMA-7B and LLaMA-2-7B models using different pruning ratios. Our experimental setup followed the configuration of LLM-Pruner, so we used only randomly selected 10 datasets for the PROScore calculation, and during the performance recovery phase using Alphaca dataset (Taori et al., 2023). For evaluation, we performed only zero-shot evaluation.

Table 4: LLAMA2-7b pruning performance comparison of various pruning method on five common reasoning datasets

| Remain Param Ratio | Method | Acc(↑%) | | | | | Avg(↑%) | Drop(↓%) |
| --- | --- | --- | --- | --- | --- | --- | --- | --- |
| | | PIQA | HellaSwag | WinoGrande | ARC-e | ARC-c | | |
| 1.0 | Baseline | 0.78 | 0.57 | 0.69 | 0.76 | 0.43 | 0.646 | 0.0 |
| 0.5 | LLM-Pruner (Ma et al., 2023) | 0.67 | 0.35 | 0.52 | 0.48 | 0.22 | 0.448 | 19.8 |
| | SliceGPT (Ashkboos et al., 2024) | 0.58 | **0.46** | **0.55** | 0.37 | **0.28** | 0.448 | 19.8 |
| | Bonsai (Dery et al., 2024) | 0.66 | 0.40 | 0.54 | 0.49 | 0.26 | 0.470 | 17.6 |
| | Wanda-sp (Sun et al., 2023) | 0.63 | 0.32 | 0.53 | 0.43 | 0.20 | 0.422 | 22.4 |
| | **IPPRO (ours)** | **0.68** | 0.43 | 0.51 | **0.50** | **0.28** | **0.480** | **16.6** |

We compared the results of the LLaMA-7B model on five common sense reasoning datasets: BoolQ (Clark et al., 2019), PIQA (Bisk et al., 2020), ARC-easy (Clark et al., 2018), ARC-challenge (Clark et al., 2018) and OpenbookQA (Mihaylov et al., 2018). As shown in Table 3, IPPRO demonstrates consistent and robust performance on all datasets. So, our average score is higher than other stat-of-the-art LLM pruning methods. Similarly, we compared the results of the LLaMA2-7B model on five common sense reasoning datasets: PIQA, HellaSwag (Zellers et al., 2019), WinoGrande (Sakaguchi et al., 2021), ARC-easy and ARC-challenge. As shown in Table 4, IPPRO does not perform well on just one specific dataset; instead, it achieves a consistently high or comparable score across all datasets. This indicates that IPPRO ensures strong generalized performance across various tasks.

## 5.4 RESULTS WITHOUT FINETUNING

To examine whether the effectiveness of IPPRO depends heavily on fine-tuning, we evaluated its performance without any fine-tuning and compared it with $L_1$-norm (He et al., 2017), Taylor (Molchanov et al., 2019), and hessian (Moosavi-Dezfooli et al., 2019) pruning methods. Using a uniform pruning ratio across all layers, IPPRO consistently outperformed all baselines on the ImageNet-1k dataset with DeiT-Tiny, as shown in Fig. 5a. Notably, when the pruning ratio is increased to an extreme level, the superiority of IPPRO becomes even more pronounced. Furthermore, even without fine-tuning, IPPRO demonstrates more stable performance in segmentation tasks using DeeplabV3-ResNet50, as shown in Fig. 5b. As a result, IPPRO demonstrate its applicability in low-resource environments such as embedded or on-device setting. Also, the results for the LLMs are in Section C.4

## 5.5 SAMPLING SENSITIVITY ANALYSIS

IPPRO computes importance scores by accumulating gradients, as defined in Eq. (4), which results in linear time complexity with respect to the number of input samples. While using the full dataset ensures maximum accuracy, it incurs substantial computational cost. To evaluate this trade-off, we analyze the sensitivity of our gradient-based importance estimation to the number of samples used. We employed the LLaMA-7B model, and our experiments showed that IPPRO is robust to subset-based sampling. As seen in Table 5, the performance difference was merely 0.04% when using

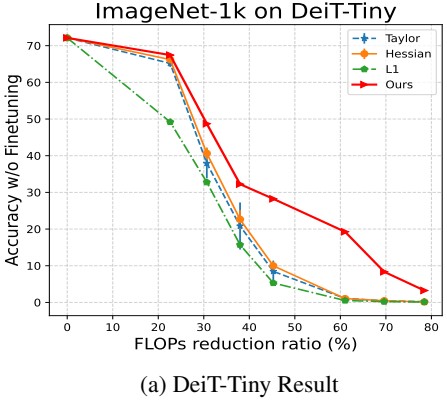
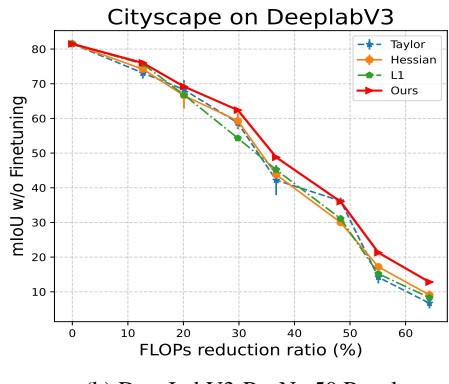

(a) DeiT-Tiny Result          (b) DeepLabV3-ResNet50 Result

Figure 5: Without Finetuning Top1 Acc and mIoU results of DeiT-Tiny and DeeplabV3 models

Table 5: LLM Pruning results on different calibration size

| Calibration Set Num samples | Method | Acc(↑%) | | | | | | | Avg(↑%) |
|---|---|---|---|---|---|---|---|---|---|
| | | BoolQ | PIQA | HellaSwag | WinoGrande | ARC-e | ARC-c | OBQA | |
| NA | Baseline | 73.1 | 78.3 | 72.9 | 66.8 | 67.3 | 41.4 | 42.4 | 63.17 |
| 10 | LLM-Pruner | 69.5 | 76.4 | **68.1** | 65.1 | 63.4 | 37.9 | 40.0 | 60.06 |
| | IPPRO (ours) | 72.8 | **76.8** | 67.5 | 65.1 | 67.2 | 37.6 | 39.6 | 60.94 |
| 100 | IPPRO (ours) | **72.9** | 75.9 | 66.7 | **66.3** | 65.6 | 37.7 | 39.0 | 60.59 |
| 1000 | IPPRO (ours) | 72.1 | **76.8** | 67.1 | 62.8 | **68.5** | **38.8** | **40.8** | **60.98** |

only 10 samples compared to 1,000. This implies an immense empirical speedup potential without sacrificing pruning performance. Additional results are available in the Section E.

## 6 Discussion

IPPRO and Catalyst regularization (Jung & Lee, 2025) are closely related, much like the relationship between magnitude pruning and Lasso regularization. As shown in Liu et al. (2017), Lasso leads to pruning decisions based on filter magnitudes after reducing the filter norm. Similarly, Catalyst regularization results in pruning decisions based on the ratio between trained filters and auxiliary parameters, which is essentially the same criterion used in IPPRO.

The "size matters" myth in pruning has proven to be a more pervasive and problematic heuristic than previously acknowledged. Although IPPRO is model- and task-agnostic, its performance is consistently competitive with, or even superior to, specialized methods such as SIRFP (Wu et al., 2025) for segmentation and SNP (Shim et al., 2024) for model-specific pruning. IPPRO's mechanism, measuring angular distance from the origin in projective space to eliminate scale dependence, is conceptually simple, yet remarkably effective. These results suggest that magnitude bias, long regarded as a minor artifact, may in fact be a central factor limiting the effectiveness of conventional pruning strategies.

The strength of IPPRO arises from the universality of PROscore. Rather than relying on filter size, PROscore defines importance through the trajectory of filters under gradient descent, ensuring that PROscore is not tied to any particular model architecture or task. Consequently, IPPRO applies consistently acriss paradigms such as CNNs, ViT, and LLMs, demonstrating broad generalization. Unlike domain-specific approaches, IPPRO establishes a truly versatile pruning paradigm grounded in magnitude-indifferent principles. As further evidenced in Section D, IPPRO is uniquely free from magnitude correlations compared to other criteria, reinforcing its position as a truly magnitude-indifferent pruning paradigm.

Despite the demonstrated strengths, IPPRO have limitations such as computational overhead facing large datasets as PROscore requires a full-batch gradient. However, as shown in Section 5.5 and Section E, PROscore remains reliable with subset-based sampling, suggesting a promising direction toward a stochastic variant for online pruning at data scale.

## 7 Conclusion

We introduce PROscore, a novel magnitude-invariant importance criterion, and bulit upon it to propose IPPRO, a structured pruning method. Unlike conventional approaches that rely on filter magnitude, PROscore leverages projective geometry to evaluate filter importance through angular displacement under gradient descent. This fundamentally challenges the longstanding "size-matters" assumption in pruning. Extensive experiments across CNNs, ViT, LLMs demonstrate that IPPRO delivers consistent and robust performance improvements regardless of model architecture or task domain. Notably, it maintains strong accuracy under high pruning ratios and limited fine-tuning scenarios. These results highlight IPPRO as a versatile and practical tool for pruning. Beyond empirical gains, our work provides the theoretical foundation of structural pruning in projective geometry, offering a fresh perspective for future pruning research. Overall, IPPRO establishes a magnitude-indifferent paradigm for importance-based pruning, advancing both the theoretical understanding and practical utility of neural network compression.

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

## LLM USAGE

We used LLMs for limited purpose for editing the manuscript only. LLMs were not used for purpose.

Table 6: Finetuning configurations for models.

| Finetuning Configs | ResNet-50 | MobileNetv2 | ResNet-56 | VGG-19 | DeiT |
|---|---|---|---|---|---|
| dataset | ImageNet-1k | ImageNet-1k | CIFAR-10 | CIFAR-100 | ImageNet-1k |
| epochs | 180 | 300 | 400 | 400 | 300 |
| batch size | 128 | 256 | 256 | 256 | 512 |
| optimizer | SGD | SGD | SGD | SGD | AdamW |
| learning rate scheduler | step | cosine | step | cosine | cosine |
| step size | 40 | - | 30 | - | - |
| base learning rate | 0.01 | 0.01 | 0.025 | 0.017 | 0.0005 |
| weight decay | 1e-4 | 1e-4 | 0.0005 | 0.005 | 0.01 |
| optimizer momentum | 0.9 | 0.9 | 0.9 | 0.9 | (0.9, 0.999) |

## A    IMPLEMENTATION DETAIL

Note that the model parameters are not updated during the PROscore computation. The extended model is reverted to the original model and then we prune the filters according to the obtained PROscore.

To reduce the randomness of the empirical results of our implementation, we fixed the pre-trained weights $W$ and the parameter $D$ obtained from the extended projective space computed using the pre-trained weights. Also, we accumulate the gradients $\nabla_{F_i}\mathcal{L}$ and $\frac{\partial\mathcal{L}}{\partial D_i}$ from each batch of the back-propagation process using the entire dataset $\mathcal{D}$, unless otherwise mentioned.

Using the accumulated gradient values, along with the original model weights and the expanded parameter $D$, the layer-wise PROscore is computed according to Eq. (4). Using the PROscore, layer-wise pruning is performed on the original base model, which does not include the extended parameter $D$ using the previously computed importance scores.

## B    FINETUNE RECIPE

In this section, we detail the hyper-parameters used for fine-tuning as shown in Table 6. All or some of the models were fine-tuned using the PyTorch 1.13 framework on a Nvidia RTX 4090 GPU. Additionally, we emphasize that we did not use external data augmentation skills such as Color-jitter or Mix-up, which often employed to improve the performance in other pruning methods, since it invokes ethical issue on fair-comparison, as pretrained model did not used them and results with additional augmentation may not show the advantage of pruning method only.

## C    ADDITIONAL PERFORMANCE ANALYSIS

### C.1    CIFAR-10 AND CIFAR-100

We evaluated IPPRO on CIFAR-10 (ResNet-56) and CIFAR-100 (VGG19) using DepGraph pre-trained weights (Fang et al., 2023). As shown in Table 7, IPPRO reduces FLOPs and parameters by up to 49% on CIFAR-10 with a 0.55% accuracy gain. Even at higher compression rates (up to 77%), accuracy drops by only  1%, outperforming other methods. On CIFAR-100, IPPRO achieves 87% FLOPs and 87% parameter reduction with minimal accuracy loss, highlighting its effectiveness for compact, high-performing models.

### C.2    IMAGENET-1K

On ImageNet-1k, we evaluate IPPRO with ResNet-50 under three compression levels (47%, 60%, and 74% FLOPs reduction). As shown in Table 8, IPPRO improves Top-1 accuracy by 0.06% at moderate pruning and remains competitive at higher compression, outperforming state-of-the-art methods. Furthermore, for other FLOP reductions on 60% and 74%, IPPRO exhibited significantly higher accuracy compared to other superior methods.

Table 7: CIFAR-10 and CIFAR-100

| Model Dataset | Method | Top-1 Acc (%) | | | Params↓ (%) | FLOPs↓ (%) |
|---|---|---|---|---|---|---|
| | | Base. | Prun. | Δ | | |
| ResNet-56 CIFAR-10 | Depgraph(Fang et al., 2023) | 93.53 | 93.77 | +0.24 | - | - |
| | HBFP(Basha et al., 2024) | 93.26 | 92.42 | -0.84 | 43.9 | 43.6 |
| | GAL(Lin et al., 2019) | 93.26 | 93.38 | +0.12 | 12.3 | 37.6 |
| | HRank(Lin et al., 2020) | 93.26 | 93.52 | +0.26 | 17.0 | 29.3 |
| | CHIP(Sui et al., 2021) | 93.26 | **94.16** | **+0.90** | 43.9 | 47.4 |
| | FPAC(Yang et al., 2023b) | 93.26 | 93.71 | +0.45 | 42.8 | 47.4 |
| | **IPPRO (ours)** | 93.53 | 94.06 | +0.53 | **49.7** | **49.3** |
| | HRank(Lin et al., 2020) | 93.26 | 90.72 | -2.54 | 68.4 | 74.1 |
| | HBFP(Basha et al., 2024) | 93.26 | 91.79 | -1.47 | 75.4 | **74.9** |
| | GAL(Lin et al., 2019) | 93.26 | 91.58 | -1.68 | 66.1 | 60.2 |
| | CHIP(Sui et al., 2021) | 93.26 | 92.05 | -1.21 | 71.8 | 72.3 |
| | SPSRC(Sun & Shi, 2024) | 93.59 | 91.65 | -1.94 | 64.7 | 63.9 |
| | **IPPRO (ours)** | 93.53 | **92.47** | **-1.06** | **77.7** | 71.8 |
| VGG19 CIFAR-100 | Depgraph(Fang et al., 2023) | 73.50 | 70.39 | -3.11 | - | 88.7 |
| | Kron-OBS(Wang et al., 2019a) | 73.34 | 60.66 | -12.6 | - | 83.5 |
| | Greg-2(Wang et al., 2020) | 74.02 | 67.75 | -6.27 | - | 88.6 |
| | EigenD(Wang et al., 2019a) | 73.34 | 65.18 | -8.16 | 90.9 | 88.6 |
| | Torque(Gupta et al., 2024b) | 73.03 | 65.87 | -7.16 | 90.8 | 88.7 |
| | GAT Transprunig(Lin et al., 2024) | 73.26 | 66.68 | -6.58 | - | **89.0** |
| | **IPPRO (ours)** | 73.50 | **70.47** | **-3.03** | 87.9 | 87.5 |

Table 8: Comparison of pruning performance on ImageNet dataset

| Model | Method | Top-1 Acc (%) | | | Top-5 Acc (%) | | | Params↓(%) | FLOPs↓(%) |
|---|---|---|---|---|---|---|---|---|---|
| | | Base. | Prun. | Δ | Base. | Prun. | Δ | | |
| ResNet-50 | SFP (He et al., 2018) | 76.15 | 74.61 | -1.54 | 92.87 | 92.06 | -0.81 | N/A | 41.8 |
| | Autopruner (Luo & Wu, 2020) | 76.15 | 74.76 | -1.39 | 92.87 | 92.15 | -0.72 | N/A | 48.7 |
| | FPGM (He et al., 2019) | 76.15 | 75.59 | -0.56 | 92.87 | 92.63 | -0.24 | 37.5 | 42.2 |
| | Taylor (Molchanov et al., 2019) | 76.18 | 74.50 | -1.68 | N/A | N/A | N/A | 44.5 | 44.9 |
| | GAL (Lin et al., 2019) | 76.15 | 71.95 | -4.20 | 92.87 | 90.94 | -1.93 | 16.9 | 43.0 |
| | HRank (Lin et al., 2020) | 76.15 | 74.98 | -1.17 | 92.87 | 92.33 | -0.54 | 36.6 | 43.7 |
| | SCOP (Tang et al., 2020) | 76.15 | 75.95 | -0.20 | 92.87 | 92.79 | -0.08 | 42.8 | 45.3 |
| | CHIP (Sui et al., 2021) | 76.15 | 76.15 | 0.00 | 92.87 | 92.91 | +0.04 | 44.2 | 48.7 |
| | RGP(Chen et al., 2023) | 76.22 | 75.30 | -0.92 | N/A | N/A | N/A | 43.8 | 43.8 |
| | FPBICI(Tang et al., 2024) | 76.13 | 76.08 | -0.05 | 92.86 | 92.85 | -0.01 | 45.9 | 50.4 |
| | **IPPRO (ours)** | 76.15 | **76.21** | **+0.06** | 92.87 | **93.02** | **+0.15** | 46.4 | 50.4 |
| | SCOP (Tang et al., 2020) | 76.15 | 75.26 | -0.89 | 92.87 | 92.53 | -0.34 | 51.8 | 54.6 |
| | SIRFP (Wu et al., 2024) | 76.15 | 75.14 | -1.01 | 92.87 | **93.12** | **+0.25** | N/A | 58.7 |
| | CHIP (Sui et al., 2021) | 76.15 | 75.26 | -0.89 | 92.87 | 92.53 | -0.34 | 56.7 | 62.8 |
| | Torque (Gupta et al., 2024a) | 76.07 | 74.58 | -1.49 | N/A | N/A | N/A | 64.5 | 57.2 |
| | FPBICI(Tang et al., 2024) | 76.13 | 75.01 | -1.12 | 92.86 | 92.30 | -0.56 | 57.7 | **63.8** |
| | **IPPRO (ours)** | 76.15 | **75.51** | **-0.64** | 92.87 | 92.67 | -0.20 | 60.0 | 60.0 |
| | HRank (Lin et al., 2020) | 76.15 | 69.10 | -7.05 | 92.87 | 89.58 | -3.29 | 67.5 | 76.0 |
| | CHIP (Sui et al., 2021) | 76.15 | 73.30 | -2.85 | 92.87 | 91.48 | -1.39 | 68.6 | 76.7 |
| | RGP (Chen et al., 2023) | 76.22 | 72.68 | -3.54 | N/A | N/A | N/A | **75.0** | 75.0 |
| | **IPPRO (ours)** | 76.15 | **73.38** | **-2.77** | 92.87 | **91.50** | **-1.37** | 74.2 | 74.2 |
| MobileNetV2 | Meta (Liu et al., 2019) | 74.70 | 68.20 | -6.50 | N/A | N/A | N/A | N/A | **54.2** |
| | GFP (Liu et al., 2021) | 75.74 | **69.16** | -6.58 | N/A | N/A | N/A | N/A | 50.5 |
| | **IPPRO (ours)** | 72.01 | 67.90 | **-4.11** | 90.62 | 88.04 | -2.58 | 42.3 | **54.2** |

We further validate IPPRO on MobileNetV2, a compact model without residual connections, where it shows the smallest accuracy drop at comparable FLOPs, demonstrating architectural versatility.

## C.3 GLOBAL PRUNING

We define our importance score using angular deviation in the projective space, measuring how much each filter responds to data relative to a reference value of $\tan(\frac{\pi}{4})$. Although the absolute score can vary with the hyperparameter $\lambda$, PROscore enables global comparability without explicit

normalization, unlike magnitude-based methods. This property is essential for global pruning, where importance scores must reside on a comparable scale to ensure fair competition among filters across layers. Without this, layers with intrinsically larger scores may be unjustly favored. To verify that score scaling induced by Eq. (4) $\lambda$ does not undermine pruning behavior, we analyze its effect and confirm the stability of our method in Section G.

Table 9: Global Pruning Performance results of ImageNet and CIFAR-10 datasets.

(a) ImageNet-1k on ResNet-50

| Method | Top-1 Acc (%) | | | Remain FLOPs(G) |
|---|---|---|---|---|
| | Base. | Prun. | Δ | |
| ResConv-Prune(Xu et al., 2020) | 76.2 | 70.0 | -6.2 | 1.6 |
| DBP-0.5(Wang et al., 2019c) | 76.2 | 72.4 | -3.8 | N/A |
| Meta(Liu et al., 2019) | 76.2 | 73.4 | -2.8 | **1.0** |
| AutoSlim(Yu & Huang, 2019) | 76.2 | 74.0 | -2.2 | **1.0** |
| GReg-2(Wang et al., 2020) | 76.2 | 73.9 | -2.3 | 1.3 |
| HALP(Shen et al., 2022) | 76.2 | 74.5 | -1.7 | 1.2 |
| **IPPRO (ours)** | 76.2 | **74.6** | **-1.6** | 1.3 |
| HALP(Shen et al., 2022) | 76.2 | 68.1 | -8.1 | 0.6 |
| MDP (Sun et al., 2024) | 76.2 | **70.0** | **-6.2** | **0.5** |
| **IPPRO (ours)** | 76.2 | 69.9 | -6.3 | 0.6 |

(b) CIFAR-10 on ResNet-56

| Method | Top-1 Acc (%) | | | Remain FLOPs(M) |
|---|---|---|---|---|
| | Base. | Prun. | Δ | |
| FSM(Duan et al., 2022) | 93.26 | 93.63 | +0.4 | 61.17 |
| ITFCP(Chen & Wang, 2024) | 93.39 | 93.60 | +0.21 | 60.73 |
| GCNNA(Jiang et al., 2022) | 93.72 | 93.72 | 0 | **58.29** |
| FSIM(Liu et al., 2023) | 93.30 | 93.48 | +0.18 | 59.24 |
| **IPPRO (ours)** | 93.53 | **94.00** | **+0.47** | 63.66 |
| QSFM(Wang et al., 2022) | 93.21 | 91.88 | -1.33 | 50.62 |
| GBN(You et al., 2019) | 93.10 | 91.76 | -1.34 | 40.23 |
| FSIM(Liu et al., 2023) | 93.30 | 91.96 | -1.34 | **31.08** |
| **IPPRO (ours)** | 93.53 | **92.43** | **-1.10** | 37.49 |

By leveraging this robust scoring scheme, our global pruning method consistently outperforms conventional magnitude-based global pruning. As shown in Table 9, it also achieves performance on par with other advanced approaches specifically designed to address the shortcomings of norm-based criteria.

## C.4 RESULTS WITHOUT FINETUNING OF LLMS

To assess the impact of fine-tuning on LLMs pruning, we evaluated IPPRO without any fine-tuning and compared it against $L_1$-norm (He et al., 2017) and Taylor (Molchanov et al., 2019) pruning methods using LLM-Pruner. We pruned the models to retain 20% and 30% of the parameters and reported the results on nine datasets, as shown in Table 10. At the 20% retention level, the performance of IPPRO was generally comparable to Taylor, with slight variations across some datasets. In contrast, at the 30% retention level, IPPRO consistently outperformed all other methods across every dataset. These results indicate that, even without fine-tuning, IPPRO provides stable and reliable performance when applied to LLMs.

Table 10: LLAMA-7b without finetune results

| Remain Param Ratio | Method | WikiText2 (↓%) | PTB (↓%) | BoolQ | PIQA | HellaSwag | WinoGrande | ARC-e | ARC-c | OBQA | Avg | Drop(↓%) |
|---|---|---|---|---|---|---|---|---|---|---|---|---|
| 1.0 | Baseline | 12.62 | 22.14 | 73.1 | 78.3 | 72.9 | 66.8 | 67.3 | 41.4 | 42.4 | 63.17 | 0.0 |
| 0.2 | LLM-Pruner (L1) | 236.23 | 446.55 | 50.52 | 57.89 | 40.42 | 51.46 | 35.86 | 27.99 | 27.80 | 41.70 | 21.47 |
| | LLM-Pruner (Taylor) | 19.77 | 36.66 | 59.39 | **75.57** | **65.34** | 61.33 | **59.18** | **37.12** | **39.80** | 56.82 | 6.35 |
| | IPPRO (ours) | **17.89** | **29.39** | **65.26** | 74.54 | 64.06 | **63.22** | 58.26 | 34.56 | 38.20 | **56.87** | **6.3** |
| 0.3 | LLM-Pruner (L1) | 294.00 | 446.55 | 41.47 | 56.58 | 35.62 | 50.91 | 32.11 | 25.51 | 29.80 | 38.86 | 24.31 |
| | LLM-Pruner (Taylor) | 32.85 | 74.33 | 62.17 | 68.77 | 59.49 | 53.28 | 45.83 | 30.38 | 35.40 | 50.76 | 12.41 |
| | IPPRO (ours) | **18.89** | **31.63** | **65.26** | **73.75** | **61.67** | **62.12** | **54.24** | **33.79** | **36.60** | **55.35** | **7.82** |

## D COMPARISON OF IMPORTANCE CRITERIA

We compare IPPRO with other magnitude- and gradient-based pruning methods such as $L_1$-norm (He et al., 2017), Taylor (Molchanov et al., 2019), and geometric (He et al., 2019). For fairness, we compute gradients across the entire dataset, even for methods like Taylor that support randomized subsets. As shown in Fig. 6, traditional methods heavily depend on pre-trained weights and produce highly similar pruning patterns. In contrast, IPPRO demonstrates reduced reliance on initial weights, yielding distinct importance distributions. Histogram visualizations of ResNet-50 layers further confirm that IPPRO prunes differently, supporting its robustness against weight initialization bias.

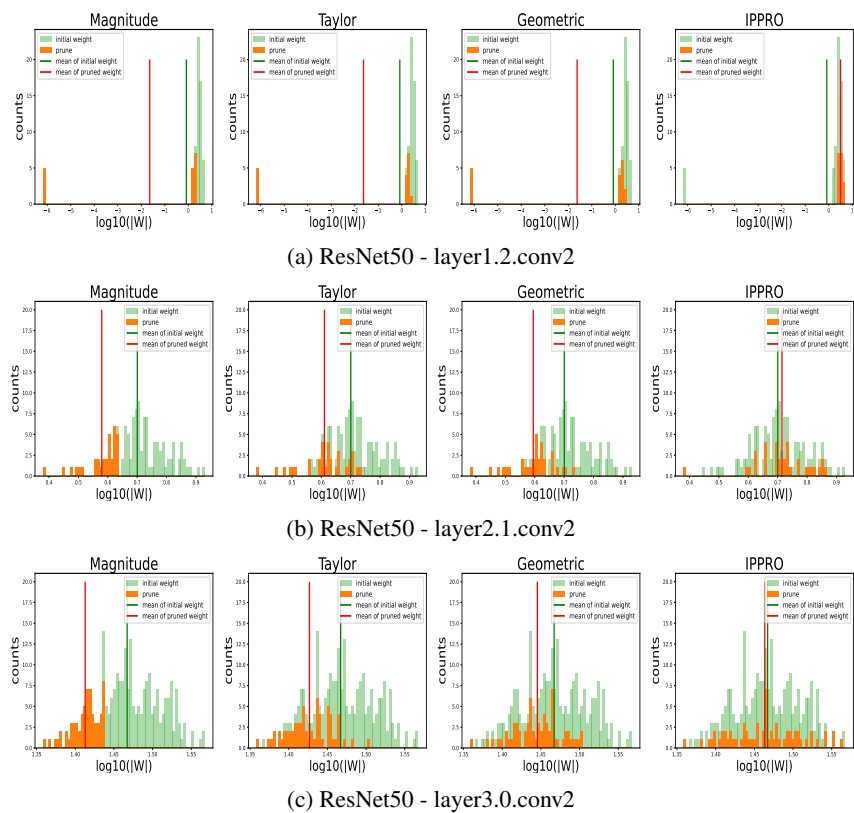

(a) ResNet50 - layer1.2.conv2

(b) ResNet50 - layer2.1.conv2

(c) ResNet50 - layer3.0.conv2

Figure 6: Visualization of magnitude of pruning filters obtained by four different criteria, on DeepLabV3-Resnet50 with Cityscapes dataset.

# E  SENSITIVITY ANALYSIS

We conduct experiments on ResNet-56 with CIFAR-10 and VGG-19 with CIFAR-100, varying the number of samples used to 100%, 50%, 25%, and 5% of the dataset. A balanced label sampler is employed to preserve label distribution across mini-batches. To account for randomness introduced by subset sampling, each configuration is repeated five times, and both the mean (Mu) and the maximum accuracy are reported.

Table 11: Experimental results on different dataset size

(a) CIFAR10 dataset on Resnet-56 (FLOPs reduction 71.85%)

| Dataset size | Time usage (s) | Top-1 Acc (%) | | |
|---|---|---|---|---|
| | | Base. | Prun. Mu (Max) | $\Delta$ Mu (Max) |
| Full | 45.4 | 93.53 | 92.47 | -1.06 |
| 50% | 25.4 | 93.53 | 92.18 (92.44) | -1.35 (-1.09) |
| 25% | 16.6 | 93.53 | 92.27 (92.46) | -1.26 (-1.07) |
| 5% | 7.2 | 93.53 | 92.21 (92.43) | -1.32 (-1.10) |

(b) CIFAR100 dataset on VGG19 (FLOPs reduction 87.5%)

| Dataset size | Time usage (s) | Top-1 Acc (%) | | |
|---|---|---|---|---|
| | | Base. | Prun. Mu (Max) | $\Delta$ Mu (Max) |
| Full | 13.2 | 73.5 | 70.47 | -3.03 |
| 50% | 6.9 | 73.5 | 69.66(70.09) | -3.84(-3.41) |
| 25% | 3.5 | 73.5 | 69.55(70.03) | -3.95(-3.47) |
| 5% | 0.7 | 73.5 | 69.58(70.24) | -3.92(-3.26) |

# F  QUALITATIVE RESULTS OF DEEPLABV3

In this section, we present qualitative results of the segmentation task. We compare the original image, ground truth segmentation, segmentation result from the unpruned model, and the segmentation result after pruning using our PROscore.

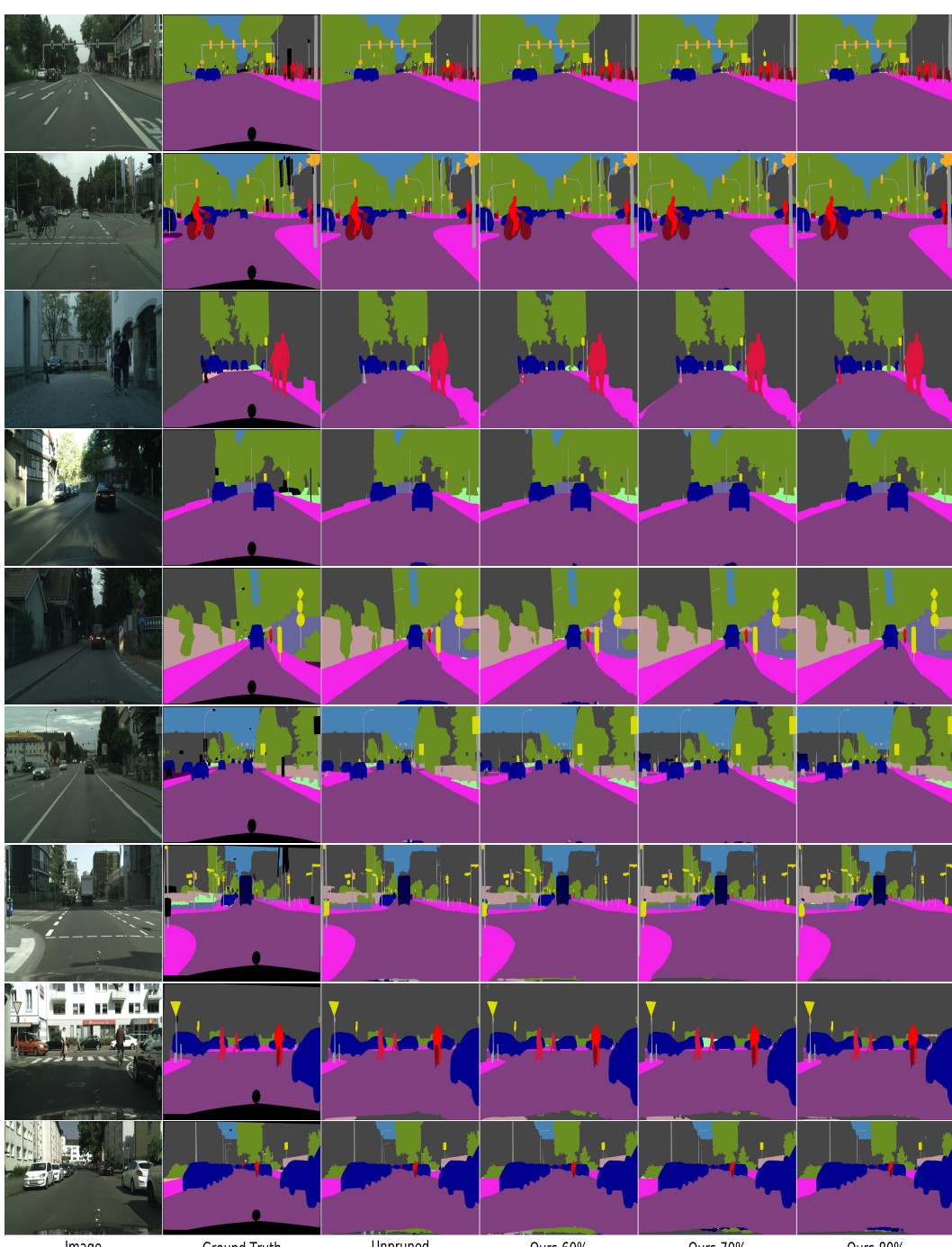

Figure 7: Visualization CitySpace dataset Our Pruning results of using DeeplabV3-ResNet50 models

## G    SENSITIVITY ANALYSIS OF $\lambda$ VALUE

The absolute value of PROscore is affected by the hyperparameter $\lambda$, which controls the step size of the gradient movement in real projective space $\mathbb{RP}^N$, since the angle $\theta(p_i')$ is related to the length $\overline{p_i p_i'}$. In extreme case, if $\lambda \to 0$ then $\theta(p_i') \to \frac{\pi}{4}$ and thus the PROscores would distribute close to one.

However, the pruning decision are invariant under selection of isotropic scaling on $\lambda > 0$, since we are choosing filters with respect to their relative order which are invariant: if the $p_i''$ is new point under gradient movement with different step size $\lambda$, then following holds: if $\theta(p_i') < \theta(p_j')$ then $\theta(p_i'') < \theta(p_j'')$.

To validate this, we conduct experiments on Resnet56 model with $\lambda$ set to 1, 0.1, 0.01, and 0.001, examining both the preservation of pruning indices and the consistency of PROscore scales across layers. As shown in Fig. 8a, the pruning indices remained stable regardless of the $\lambda$ value, and Fig. 8b shows that the relative importance measured by PROscore is invariant under $\lambda$ selection.

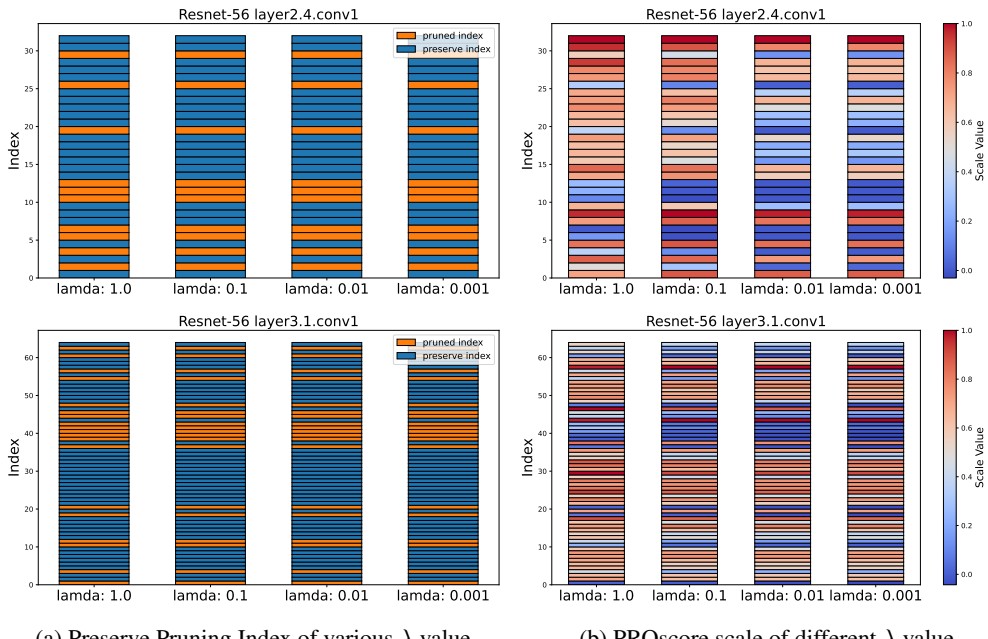

(a) Preserve Pruning Index of various $\lambda$ value        (b) PROscore scale of different $\lambda$ value

Figure 8: Preserve pruning index of $\lambda$ changes

