# OpenReview forum: "IPPRO: Importance-based Pruning with PRojective Offset for Magnitude-indifferent Structural Pruning"
_ICLR.cc/2026/Conference — Submitted to ICLR 2026_

### Official Review · Reviewer_8L8K · 2025-10-31

**Soundness:** 2
**Presentation:** 3
**Contribution:** 2
**Rating:** 2
**Confidence:** 4

**Summary:**

This paper proposes IPPRO (Importance-based Pruning with PRojective Offset), a magnitude-indifferent structured pruning method for neural networks. The key innovation lies in using projective geometry to evaluate filter importance through angular displacement under gradient descent, rather than relying on filter magnitudes.

The authors introduce PROscore, computed as the tangent of angular distance in projective space after one gradient descent step, as the importance criterion. Extensive experiments demonstrate IPPRO's effectiveness across CNNs, Vision Transformers, and LLMs, showing consistent performance improvements especially in high compression scenarios and without fine-tuning.

**Strengths:**

1. The projective geometry framework provides a principled alternative to magnitude-based heuristics.

2. Testing on 7+ architectures (ResNet, MobileNet, DeiT, EfficientFormer, DeepLabV3, LLaMA).

3. Robust performance without fine-tuning.

**Weaknesses:**

1. Why should angular distance in projective space correspond to filter importance? Is there a theoretical justification that angular distance is better than geometric distance or norm? A formal theorem would strengthen the contribution. It would also be important to theoretically justify why angular distance is superior, rather than merely demonstrating that it performs better.

2. No wall-clock time comparisons with baselines. This is important for pruning papers.

3. The performance improvement is marginal. For example, in Table 2 (Cityscapes – DeepLabV3-ResNet50), when comparing IPPRO (ours) with SIRFP (Wu et al., 2025), the improvement in mIoU is very small. Under the base pruning setting, SIRFP achieves 81.3 mIoU, while IPPRO reaches 81.5 mIoU, giving only a +0.2 gain. In terms of FLOPs reduction, IPPRO obtains 61.8 %, just 0.5 % higher than SIRFP’s 61.3 %. This slight edge shows that the performance improvement is marginal, i.e., the proposed method performs almost on par with the previous state-of-the-art under the same compression ratio.

4. No analysis on modern architectures (e.g., Swin Transformer).

5. Experiments on Imagenet should be put in the main text, not supplementary.

**Questions:**

see weaknesses

---

> ### Author Response · Authors · 2025-11-24
>
> Dear reviewer 8L8K,
> We are grateful for the thorough review. Below, we clarify the key points raised:
>
> > Why should angular distance in projective space correspond to filter importance?
> >
>
> In practice, magnitude-invariant comparisons are usually achieved by normalizing filters and applying cosine similarity. However, pruning cannot rely on such normalization because filters near zero make normalization undefined, creating an inherent singularity that prevents a continuous importance measure from being defined.
>
> Projective space provides the minimal extension needed to remove this discontinuity. By introducing a single additional homogeneous coordinate, it gives a well-defined geometric representation even for filters with vanishing magnitude. This allows stable, magnitude-invariant comparisons without requiring explicit normalization. PROscore simply measures whether a filter’s one-step gradient update moves toward or away from the origin in this projective space. Thus, angular displacement is not a heuristic but a principled importance signal derived from the geometry itself.
>
> Prior theory [link](https://arxiv.org/abs/2507.14170) further supports the logic of IPPRO, that the training dynamics on minimization of lossless pruning manifold, the $V(DW)$, separates pruned/preserved filters by pushing PROscore toward 0 or infinity. Hence, the dynamics toward manifold of lossless pruning, which minimizes the norm of defining eqation $\|DW\|$, prefers the filters with small PROscore to prune after regularization.
>
> > No wall-clock time comparisons with baselines. This is important for pruning papers.
> >
>
> Thank you for raising this point. In pruning work, computational efficiency is typically reported using FLOPs or measured latency. Since we already provide FLOPs reduction, a 60% drop directly implies roughly a 1.67× speedup. That said, we agree that including explicit latency measurements would make the evaluation clearer.
>
> To address this, we will add a latency comparison table in the revised version. We have already measured inference latency under consistent hardware and batch-size settings, and we will include the following results:
>
> - The LLAMA/LLAMA2 latency is tested under the test set of WikiText2 on a single A100 GPU.
> - The DeiT latency is benchmarked with 200 warm-up runs and averaged over 1,000 runs. The measurement is performed on an RTX 4090 GPU with a batch size of 64.
>
> |Model|baseline latency|pruned latency|baseline params| pruned params|
> |--|--|--|--|--|
> |Deit-Tiny|15.89 | 9.93 |5.7 M |3.3 M|
> |Deit-Small| 39.63| 29.34 | 22.1 M |10.7 M|
> |LLAMA 7B| 43.27| 36.34 |6.74 B|5.12 B|
> |LLAMA2 7B| 42.60| 20.92|6.74 B|3.28 B|
>
>
> > The performance improvement is marginal.
> >
> We do not consider this outcome a limitation. Matching a specialized method with a general one is a feature, not a bug. SIRFP is a segmentation-specific method, and even within its own domain the reported gains over prior SOTA are relatively modest. In contrast, IPPRO is completely task-agnostic and architecture-agnostic, yet still matches or surpasses SIRFP on Cityscapes.
>
> More importantly, the contribution of IPPRO is not a marginal, incremental enhancement over prior work, but a methodological innovation that introduces an entirely new geometric formulation for structured pruning. Unlike existing pruning methods that iteratively accumulate minor improvements on magnitude-based heuristics, IPPRO is the first to establish importance-based pruning in projective space and to remove magnitude dependence altogether. This conceptual shift is substantially more impactful than a small numerical improvement on a single benchmark, and it provides a new theoretical foundation for future research.
>
> IPPRO achieves state-of-the-art results across a broad range of tasks. In particular, on DeiT, IPPRO outperforms methods such as SNP (ECCV 2024). Additionally, as can be seen from the table, our method exhibits clear strengths even when compared to the most recent work, VBP (ICCV 2025, Oral). IPPRO also performs strongly on LLaMA, underscoring its applicability across very different architectures.
>
> |Method|MACs(G)|speedup|baseline acc|w/o finetune|
> |--|--|--|--|--|
> |DeiT-T|1.26|1.00|-|-|
> |VBP (ICCV 2025, Oral)|0.91|1.38|72.20|39.58|
> |IPPRO |0.91|1.37|72.20|44.337
> |DeiT-S|4.61|1.00|-|-|
> |VBP (ICCV 2025, Oral)|3.21|1.43|79.70|64.44|
> |IPPRO |3.20|1.44|79.70|65.13|
>
> From this perspective, the Cityscapes results should not be interpreted as marginal improvements; rather, they demonstrate that a general-purpose, theoretically novel pruning method can remain competitive with domain-specialized techniques, which highlights the strength and generality of our approach.

---

> > ### Author Response · Authors · 2025-11-24
> >
> > > No analysis on modern architectures (e.g., Swin Transformer).
> > >
> >
> > Our evaluation strategy prioritized breadth across distinct architectural paradigms (CNNs, ViTs, and LLMs) rather than exhaustively cataloging specific variants. Although Swin Transformer is not included, we do evaluate on LLaMA2, which represents a more recent and substantially more complex regime than Swin Transformer. Given this, we consider our results on LLaMA2 sufficient to demonstrate generalized performance.
> >
> > > Experiments on Imagenet should be put in the main text, not supplementary.
> > >
> >
> > We recognize ImageNet results are important. Our decision to place them in the appendix was intentional: the CNN pruning benchmark currently suffers from a consistency issue. CIFAR models lack standardized pretrained checkpoints, and many ResNet-50 ImageNet pruning papers rely on additional “good tricks” (e.g., stronger augmentations, modified training recipes) that differ from the official pretrained model. These inconsistencies make direct comparison difficult and, in some cases, potentially misleading.
> > To ensure strictly fair comparisons, we adhered to the standard pre-trained recipe. However, we understand the preference for visibility and are happy to move key ResNet-50 results to the main text in the final revision.

---

> ### Comment · Reviewer_8L8K · 2025-11-26
>
> Thanks for the response.
>
> 1. The authors mentioned the paper [a] for theoretical justification. Could you clarify whether you directly use the idea from paper [a] or build on top of this? If built on top of this paper [a], what is your unique theoretical contribution?
>
> > More importantly, the contribution of IPPRO is not a marginal, incremental enhancement over prior work, but a methodological innovation that introduces an entirely new geometric formulation for structured pruning.
>
> 2. Some previous papers, like [b], also use the geometric formulation for structured pruning. So I believe it's not the first work to use the geometric formulation.
>
> 3. For the comparison table with DeiT-T and VBP (ICCV 2025, Oral), please add more recent (year 2025) papers and include the **accuracy with finetuning**.
>
>
> [a] Catalyst: a Novel Regularizer for Structured Pruning with Auxiliary Extension of Parameter Space
>
> [b] Z.-S. Huang and C.-p. Lee, “Training structured neural networks through manifold identification and variance reduction,” in Proc. Int. Conf. Learn. Represent., 2022.

---

> > ### Author Response · Authors · 2025-12-03
> >
> > >The authors mentioned the paper [a] for theoretical justification. Could you clarify whether you directly use the idea from paper [a] or build on top of this? If built on top of this paper [a], what is your unique theoretical contribution?
> > >
> >
> > Catalyst demonstrates, through an algebraic analysis of the  minimization problem, that all critical points are global minima. Building on this, the paper shows that the gradient-dynamics ratio $c=D_ii / ||F_i||$ undergoes a bifurcation driven by its movement along the lossless-pruning manifold, and leverages this phenomenon to construct a regularization-based pruning method.
> >
> > When examining how pruning evolves in Catalyst, one can observe that this gradient-dynamics ratio is structurally similar to the initial projective distance used in IPPRO, $tan(\theta(p_i)=||F_i||/D_i$, which provides a theoretical justification for why IPPRO’s angle-based metric can serve as a meaningful measure of filter importance.
> >
> > However, Catalyst operates only when a specific regularization term is applied during training and requires optimization. In contrast, IPPRO introduces projective space into pruning for the first time, enabling Catalyst’s bifurcation insight to manifest in a far simpler and more direct theoretical formulation. IPPRO achieves scale invariance without additional parameters by embedding each filter $F_i$ into homogeneous coordinates, $[||Fi|| : Fi]$, thereby placing the model in a projective manifold. We then determine whether a gradient step moves the filter toward the origin in projective space using the updated projective angle $tan(\theta(p_i '))=||F_i-\lambda\nabla_{F_i}\mathcal{L}|| / |D_i-\lambda\frac{\partial \mathcal{L}}{\partial D_i}|$, This angle-to-zero perspective naturally eliminates the singularity issues that plague magnitude-based pruning when $||F_i||$ is near zero.
> >
> > In summary, the unique theoretical contributions of IPPRO are as follows:
> >
> > 1. We introduce projective space into the pruning literature for the first time by embedding each filter $F_i$ into homogeneous coordinates.
> >
> > 2. By operating in projective space, we resolve the normalization singularity that commonly arises in magnitude-based pruning methods when filter norms approach zero.
> >
> > 3. We propose a new perspective for defining filter importance by examining changes in angular distance induced by a gradient step in projective space.
> >
> > >Some previous papers, like [b], also use the geometric formulation for structured pruning. So I believe it's not the first work to use the geometric formulation.
> > >
> >
> > Paper (b) aims to train structured neural networks using a Group-Lasso regularizer. The geometric arguments in (b) refer to optimization geometry, specifically the identification of an active manifold induced by partly-smooth regularizers, rather than any geometric formulation for pruning. Importantly, (b) explicitly distinguishes its setting from pruning. As stated on page 2: “However, as a post-processing approach, pruning is essentially different from structured training considered in this work…”. Although pruning is briefly mentioned in (b), its treatment is entirely limited to the use of Group Lasso and does not involve any geometric pruning mechanism. There is no projective formulation, no angular analysis, and no scale-invariant filter-level geometry in (b).
> >
> > In contrast, IPPRO is a pruning algorithm, not a regularized training method. Our geometric formulation is fundamentally different:
> >
> > 1. We introduce projective space into pruning for the first time by embedding each filter into homogeneous coordinates and analyzing its angular motion under gradient steps.
> >
> > 2. Our geometry is projective space, not partly smooth optimization geometry.
> >
> > 3. The resulting PROscore provides a scale-invariant, magnitude-indifferent filter-importance metric, none of which appears in (b).
> >
> > (b) does not discuss projective embeddings, does not analyze angular distance to the origin, and does not provide any geometric pruning rule.
> >
> > Therefore, (b) is conceptually and technically orthogonal to our contribution. IPPRO is the first work to introduce a projective-space based geometric embedding for structured pruning, enabling a new class of magnitude-independent pruning criteria that are absent in prior work, including (b).

---

> > > ### Author Response · Authors · 2025-12-03
> > >
> > > >For the comparison table with DeiT-T and VBP (ICCV 2025, Oral), please add more recent (year 2025) papers and include the accuracy with finetuning.
> > > >
> > >
> > > We intentionally reported VBP separately because its evaluation protocol differs substantially from the original DeiT-Tiny training recipe. Specifically, VBP uses a distillation-based training scheme with an additional 10-epoch finetuning stage, whereas the standard DeiT-Tiny baseline is trained without distillation for 300 epochs. This mismatch makes a direct, table-level comparison with other methods unfair.
> > >
> > > Regarding other recent works (2025), most of them target different sparsification objectives, such as token pruning or trajectory-based sparsification, rather than structural pruning comparable to ours. For this reason, we did not include them in the comparison table.
> > >
> > > Nevertheless, even under VBP’s own finetuning setting, IPPRO achieves higher accuracy; and under the standard DeiT training regime, the performance gap becomes even larger.
> > >
> > > |Method|MACs(G)|speedup|baseline acc|w/o finetune|w/ finetune|
> > > |--|--|--|--|--|--|
> > > |DeiT-T|1.26|1.00|-|-|-|
> > > |VBP (ICCV 2025, Oral)|0.91|1.38|72.20|39.58|70.61|
> > > |IPPRO |0.91|1.38|72.20|44.34|70.84|

---

### Official Review · Reviewer_VazZ · 2025-10-31

**Soundness:** 4
**Presentation:** 4
**Contribution:** 2
**Rating:** 6
**Confidence:** 4

**Summary:**

This paper proposes IPPRO (Importance-based Pruning with PRojective Offset), a structured pruning framework that overcomes the limitations of magnitude-based pruning by focusing on directional importance. The method embeds each filter into a projective space, where importance depends on direction rather than scale, ensuring scale-invariant comparison across layers. A new PROscore measures how much each filter’s direction changes under a gradient step, and filters that move closer to the origin are pruned. A simple parameter injection trick allows this computation without affecting model outputs. Experiments on CNNs, Vision Transformers, and LLMs show that IPPRO achieves slightly better accuracy and greater stability than previous direction-based pruning methods even without fine-tuning.

**Strengths:**

**1. Clear and principled formulation of scale invariance through projective geometry**

The paper formalizes magnitude-independent pruning not as heuristic normalization but as a property of the underlying space. By embedding filters into a projective space, the method guarantees scale invariance at the definition level, providing a clean and mathematically grounded justification for direction-based importance.

**2. Unified pruning framework applicable across architectures**

The same projective formulation applies consistently to CNNs, Vision Transformers, and large language models. This unification makes the approach architecture-agnostic and highlights that the proposed importance measure is not tied to a specific model design or normalization scheme.

**3. Comprehensive empirical validation across diverse tasks**

Experiments cover image classification, semantic segmentation, and language modeling, using models such as ResNet-50, DeepLabV3, DeiT, and LLaMA-2-7B. The breadth of evaluation supports the claim that IPPRO is a general framework rather than a task-specific trick.

**4. Robustness and fine-tuning-free performance**

IPPRO maintains accuracy even when computed with limited data sampling and performs competitively without any fine-tuning after pruning. This property makes the method practical for large-scale models where retraining is expensive or infeasible.

**5. High clarity and reproducibility**

The paper is well-organized, with precise notation, clear pseudo-code, and detailed ablations that enhance transparency. The paper’s presentation quality and completeness make reproduction straightforward and strengthen the empirical credibility of the results.

**Weaknesses:**

**1. Direction-based pruning has been extensively explored in prior work**

Several recent methods such as Torque, Catalyst, and geometric pruning already focus on gradient direction rather than magnitude. IPPRO provides a cleaner mathematical reformulation but does not introduce a fundamentally new optimization insight.

**2. Limited theoretical gain from adopting projective geometry**

Although the paper frames pruning in the language of projective geometry, the practical effect largely reduces to normalizing vectors and measuring angular displacement. The framework adds elegant terminology but yields little new theoretical understanding beyond existing direction-based normalization schemes.

**3. Increased computational overhead with limited performance improvement**

Computing PROscores requires additional gradient accumulation and parameter injection, which significantly increases pruning cost. However, the resulting accuracy gain over previous direction-based methods is minimal, raising concerns about the efficiency–benefit trade-off.

**4. Unclear behavior in hybrid architectures without manual layer-wise control**

While the method is claimed to be unified across CNNs and Transformers, it is unclear how well the approach generalizes to mixed architectures such as ConViT or hybrid CNN–ViT models when global pruning is applied without manually setting layer-wise ratios. This raises questions about the true level of architectural unification achieved by the framework.

**Questions:**

See the weaknesses.

---

> ### Author Response · Authors · 2025-11-24
>
> Dear reviewer VazZ,
> We appreciate the detailed and insightful review. We have carefully considered each point and provide our responses as follows:
>
> > 1. Direction-based pruning has been extensively explored in prior work
> >
> While directional signals are indeed established in pruning, IPPRO fundamentally diverges by formulating importance in projective space. Our criterion is fundamentally rooted in projective space, where we measure a filter’s geometric projective angular distance rather than raw gradient direction. This perspective, inspired by Catalyst, provides a magnitude-invariant and well-defined notion of proximity to zero that existing direction-based methods do not capture.
>
> While methods such as Torque or geometric pruning also consider directional information, none of them adopt this projective formulation or demonstrate its effectiveness across architectures as diverse as CNNs, ViTs, and LLMs. In this sense, IPPRO is the first to show that a projective distance–based pruning principle can generalize broadly and consistently across modern models.
>
> > 2. Limited theoretical gain from adopting projective geometry
> >
>
> The theoretical contribution is both substantial and necessary. Standard normalization methods inherently suffer from a singularity as $||F|| \rightarrow 0$, rendering importance scores unstable. Projective geometry is the specific mathematical remedy for this issue.Through homogeneous coordinates, we establish a rigorously defined metric that is intrinsically scale-invariant. This yields a stable, magnitude-independent importance measure that ordinary Euclidean approaches cannot provide. This is not merely a theoretical exercise; Figure 1 demonstrates the empirical consequence. The fact that IPPRO maintains performance where standard Magnitude Pruning fails (e.g., under BatchNorm scaling) serves as definitive proof that this geometric stability is critical for robust pruning
>
> > 3. Increased computational overhead with limited performance improvement
> >
>
> While the approach does introduce a small amount of extra computation, the computational overhead of PROscore is negligible in practice. The only additional parameters, $diag(D)$, scale linearly with channels, resulting in virtually no memory footprint. Although we perform a single epoch of gradient accumulation, this cost is trivial compared to full fine-tuning. In addition, Section 5.5 and Appendix E show that using only a small subset of calibration data yields almost identical results, which further reduces the practical burden. In effect, the computational profile is very close to that of Taylor-based pruning.
>
> For the LLaMA experiments in Section 5.3, we strictly matched the calibration setup used by LLM-Pruner to ensure a fair comparison. Under the same setting, IPPRO consistently outperformed LLM-Pruner, suggesting that a sufficiently informative data subset is sufficient for estimating filter importance.
>
> Finally, compared to methods such as Torque, IPPRO delivers more stable and stronger performance across a broader set of architectures, while adding only minimal overhead.
>
>
> > 4. Unclear behavior in hybrid architectures without manual layer-wise control
> >
>
> The concern is understandable. Prior work such as NViT (ICLR 2023) shows that many pruning frameworks handle diverse architectures by combining layer-wise importance scores with additional constraints (e.g., latency). Even magnitude-based approaches, despite large layer-to-layer variance, can perform global pruning with proper calibration.
>
> IPPRO can be integrated in a similar way. PROscore is fully parallelizable and can be coupled with latency-aware or layer-balancing schemes without difficulty. Moreover, IPPRO has already demonstrated strong architecture-agnostic behavior across CNNs, ViTs, and LLMs, suggesting that extending it to hybrid architectures is technically feasible and likely to work well. However, the goal of this paper is to evaluate how well the projective-space PROscore works on its own, without adding latency models or layer-specific heuristics.
>
> Extending IPPRO toward hybrid, latency-aware global pruning is indeed valuable, but such an extension involves a broad design space and requires substantial additional experiments. Given the page constraints of the current submission, we consider it more appropriate to leave this direction for future work.

---

### Official Review · Reviewer_z984 · 2025-11-05

**Soundness:** 2
**Presentation:** 1
**Contribution:** 2
**Rating:** 4
**Confidence:** 4

**Summary:**

The authors develop an architecture-agnostic pruning algorithm (although it is not entirely clear how it would work on a simple MLP) that is based on ideas from projective geometry. Concretely, the method uses a score called PROscore (defined in Eq (4)) involving a modified version of the model under consideration that includes some fresh parameters (parameter injection, the meaning of which isn’t entirely clear) and the gradient of the loss function of this modified model.

The authors then explain how to use this score function for pruning in the case of CNNs, vision transformers and LLMs and proceed to show that the method works well in a very wide range of experiments.

**Strengths:**

- Based on the empirical section of the paper, the method clearly works very well.
- The experiments and results are convincing and well put together.
- The scoring function (4) is interesting.

**Weaknesses:**

- The paper starts from the following axiom: when it comes to pruning a NN, the idea that the magnitude of the parameters of the NN is important is a myth (143-145), and “magnitude-invariant” methods must be developed. There is a mathematical and a motivational problem with this perspective:
	- Mathematical: What operation are involved in a forward pass through a NN? Lots of additions and multiplications, ReLU and softmax to name the most important. Addition, multiplication by positive numbers, ReLU and softmax are isotone in their arguments, multiplication by negative numbers is antitone; in particular they are all monotone. Therefore, inputs with bigger magnitudes (i.e. absolute values) produce outputs with bigger magnitudes. Thus magnitude matters at a very fundamental level in any NN. The onus is therefore on the authors to substantiate their claim that the importance of magnitude is a myth on the face of this very basic mathematical fact. Which brings me to the second point,
	- Motivational: one cannot do science by simply stating that a certain way of doing things is a myth and taking another approach. If magnitude is not, or less, important than one might think, this must be documented and explained. This paper doesn't do either.
- Despite the claim that the proposed method is “magnitude-invariant” (a term that is never defined precisely), the proposed solution is very much magnitude dependent. The expression (4) -- which has a typo, it should $\lVert F_i\rVert$ in the denominator, not $D_i$ -- is large when the numerator $\lVert F_i-\lambda\nabla_{F_i}\mathcal{L} \rVert$ is large, i.e. the updated $F_i$ with learning rate $\lambda$ has a large magnitude, and the denominator $\lvert \lVert F_i\rVert-\lambda\frac{\partial\mathcal{L}}{\partial D_i} \rvert$ is small (the interpretation of this term is more complicated, see below). In which sense is this magnitude-invariant?
- The expression (4) could have been written without any reference to projective space and projective geometry. It really plays no role in this story. Eq (4) provides a gradient-based method with an unusual denominator, and this denominator is the heart of the story.
- I wish more time and care had been spent on this denominator $\lvert \lVert F_i\rVert-\lambda\frac{\partial\mathcal{L}}{\partial D_i} \rvert$. To start to understand it one must jump to the next section to find the definition of $\frac{\partial\mathcal{L}}{\partial D_i}$ (this is not a great way to present things…). But this is not enough to understand the meaning of this term. What exactly is $\psi$ in (5)? What does “modifying element-wise computation layer $\sigma$” mean? If $\sigma(x)$ returns a tuple of dimension N and $x$ is a tuple of dimension $M\neq N$ how are we supposed to understand (5)? Since $\frac{\partial\psi}{\partial D_i}=x$ (and the second derivative will therefore vanish), what can we say about the general shape of $\frac{\partial\mathcal{L}}{\partial D_i}$ from the chain rule? And what does this mean for the proposed method and the meaning of the denominator of (4)?
- The paper is littered with grammatical errors, making it quite hard to read in some places.

**Questions:**

If the proposed method does work better than other pruning strategies, it looks like it is due to the denominator in (4) which penalises certain behaviours. Which behaviours and how? What is the role of $\lambda$ (and how is it chosen)? What is the intuition behind $\frac{\partial\mathcal{L}}{\partial D_i}$? What rate of change does it measure?

---

> ### Author Response · Authors · 2025-11-24
>
> Dear reviewer z984,
> Thank you for the detailed review. It seems that some aspects of the projective-space construction and the update rule may have caused confusion due to our limited explanations. We clarify these mechanics in detail below:
>
> > “magnitude-invariant” mathematical problem
>
> To clarify, we do not dismiss the utility of magnitude-based priors; they are indeed simple and effective baselines. However, our focus is on addressing the structural instabilities where magnitude relies fail. Specifically, magnitude metrics become unreliable because they:
>
> The point we raise is that magnitude-driven criteria can become unreliable under certain conditions. In particular:
>
> - magnitude pruning behave inconsistently across different normalization schemes,
> - under high compression, only large-magnitude filters remain, causing performance collapse,
> - magnitude-based criteria (e.g., Taylor) remain strongly correlated with magnitude.
>
> IPPRO is proposed as an alternative importance measure for precisely these scenarios, and Figures 1–2 provide empirical evidence supporting this motivation.
>
>
> > motivational problem
>
> The fact that magnitude-based methods often fail indicates the need for a mathematically well-defined approach that does not rely on the heuristic assumption of ‘smaller-norm-less-important.’ In this context, gradient-direction pruning has been developed, the Catalyst [link](https://arxiv.org/abs/2507.14170) paper in particular already incorporates theoretical elements that align closely with our projective-space perspective. Our contribution is to generalize the advantages of projective space and reformulate the theory in a way that makes it more broadly applicable and easier to use across diverse tasks and architectures. As a result, compared with recent methods SNP(ECCV 2024) , our approach achieves state-of-the-art performance within the DeiT model and delivers competitive results on LLAMA-based models as well. Furthermore, even on segmentation tasks where task-specific methods like SIRFP are specifically designed to excel our method matches or surpasses their performance. We believe this demonstrates that the mathematical properties of projective space provide meaningful benefits for determining pruning targets. Although we do not explore projective space in full depth, we explicitly acknowledge Catalyst as a key theoretical motivation for our formulation.
>
>
> > typo error, $||F_i||$ should in the denominator, not $D_i$ in eq (4)
>
> As noted in Section 3.2 (line 194), we define $||F_i|| = D_i$, so Eq. (4) is consistent with this notation and is not incorrect.
> We will clarify this correspondence more explicitly in the revised version to avoid confusion.
>
> > The expression (4) could have been written without any reference to projective space and projective geometry. It really plays no role in this story.
>
> Sections 3.1–3.2 rigorously define two essential components:
>  $embed(F_i) = [||F_i||: F_{i1}: \cdots: F_{iN}]$ satisfies $embed(cF) = embed(F)$ for $c>0$, meaning it removes magnitude entirely and preserves only direction.
>
> This removes the magnitude information while preserving only the directional component of each filter.
> This operation can be understood as adding a projective axis to the original parameter space $\mathbb{R}^N$ and mapping it into the projective space $\mathbb{RP}^N$.
>
> A crucial consequence is that all filters are mapped to the same angular distance $\pi/4$ from the origin, which cannot be achieved by simple normalization or rescaling in $\mathbb{R}^N$.
> This property is inherent to projective geometry and forms the theoretical basis of our “magnitude-invariant’’ criterion, consistent with the theoretical work we cited previously.
>
> starting from the embedded initial point $[||F_i|| : F_i]$, the gradient update produces the expression in Eq.(3), which can be rewritten using  $D_i = ||F_i||$ as  $\left[D_i-\lambda\frac{\partial \mathcal{L}}{\partial D_i}:F_i-\lambda\nabla_{F_i}\mathcal{L}\right]$.
>
> Our aim is to quantify how this update changes the filter’s direction. To express this directional displacement numerically and use it as an importance score, we introduce the tangent expression, which leads directly to Eq. (4). Thus, the reference to projective geometry is not cosmetic; it is the mathematical structure that enables a well-defined, magnitude-invariant notion of angular movement.

---

> > ### Author Response · Authors · 2025-11-24
> >
> > >  And what does this mean for the proposed method and the meaning of the denominator of (4)?
> >
> > In our formulation, $D_i$ serves as an auxiliary parameter specifically introduced to construct the projective embedding. Which effectively expands the original parameter space. Once this extended space is defined, we measure how the gradient update changes the angular distance, and this requires looking at the update of both the original filter $F_i$ and the auxiliary parameter $D_i$ . As shown in Figure 3, $|D_i|$ and $||F_i||$ together determine how the point on the $tan(\pi/4)$ axis shifts after the update. Consequently, assigning a standalone meaning to the denominator is geometrically invalid. The denominator is an integral component of this projective rotation, inseparable from the numerator."
> >
> > > What exactly is $\psi$in (5)?
> >
> > $\psi$ is the parameter-injection function defined as $\psi(x) = Dx - \overline{D}x + \sigma(x)$, where $||F||=D$  and $D = \overline{D}$ at initialization. This ensures that adding auxiliary parameters for projective embedding does not change the model’s forward output because $ Dx - \overline{D}x = 0$. It is a trick to preserve the baseline model output.
> >
> > > What does “modifying element-wise computation layer " $\sigma$ mean?
> >
> > $\sigma$ refers to the original element-wise operation in the architecture, such as BN scaling parameters or ReLU activation, depending on what is being pruned.
> >
> > > The paper is littered with grammatical errors, making it quite hard to read in some places.
> >
> > We appreciate the comment and apologize for any sections that were difficult to follow. We will thoroughly revise the manuscript for clarity and correctness, and ensure that the final version meets a higher standard of readability.
> >
> > > What is the role of $\lambda$ (and how is it chosen)?
> >
> > $\lambda$ is a hyperparameter, and we simply used  $\lambda=1$ for all experiments.
> > Although one may worry that large gradients relative to filter magnitude may alter pruning decisions, Appendix F shows that the decisions remain stable.

---

### Official Review · Reviewer_4m1G · 2025-11-09

**Soundness:** 1
**Presentation:** 1
**Contribution:** 1
**Rating:** 0
**Confidence:** 5

**Summary:**

The work introduces a structured pruning approach for neural nets, that uses projective space. This attempts to address some of the shortcomings of magnitude based pruning. It projects the filters on latent space and then notices its movement during gradient descent. An importance score is then computed leveraging the movement information, which in turn forms the basis for pruning the filters.

**Strengths:**

The only strength of the article, in my opinion, is that it attempts to address a very timely problem is neural net.

**Weaknesses:**

The papers has several major weaknesses.

1. First and foremost, the description of the proposed method is very poor. It is very difficult to understand what is going on. I do not think a reader would be able to implement the algorithm from the description given in the paper. Specifically, I do not find the following critical information.
  A) When does it stop taking the filters out from the net?
  B) What happens if all filters are removed from a layer and how does the approach handle layer collapse?
  C) Does it need pre-trained model always?

2. Projective geometry seems the key idea of the paper. Yet, there is hardly any clarity in the paper why and how does it help. The description seems too superficial and cursory.

3. Algorithm 1 does not add any value, in my opinion. Rather the authors should use the space to better justify why and how of projective geometry.

4. Results: Finally, the results are nowhere close to the state of the art. For example, on CIFAR 10, CURL (Neural network pruning with residual-connections and limited-data, 2020), SPvR (SPvR: Structured Pruning via Ranking , 2025), Hrank ( Hrank:Filter pruning using high-rank feature map, 2020), OTOv2 achieves the same performance as the proposed method with 40% (absolute) lesser parameters. Overall, I think the proposed approach uses much more parameters than the state of the art models to achieve the same performance.

**Questions:**

I have no additional question. In weakness section, I have detailed the issues.

---

> ### Author Response · Authors · 2025-11-24
>
> We are concerned that parts of our paper may have been misunderstood, and thus we would like to provide a brief summary before proceeding.
>
> ### Summary
> Magnitude-based pruning is powerful and simple, but as illustrated in Figure 1, it still breaks down under several scenarios.
> These limitations have motivated many follow-up approaches that modify or compensate for magnitude in various ways. However, Figure 2 indicates that most of these approaches remain strongly correlated with magnitude and ultimately continue to rely on it, revealing a fundamental structural limitation. To move beyond this limitation, we argue that a mathematically grounded alternative framework is required. That entirely removes the reliance on magnitude. Accordingly, Section 3.1 explains how filters are embedded into projective space, Section 3.2 details how importance is defined within this space, and Section 4 describes how this embedding is implemented in practice.
>
> 1. Scale invariance ensured by mathematical definition.
>
> Existing direction-based pruning methods rely on implicit or heuristic normalization, but such normalization becomes undefined for zero or near-zero filters, creating a singularity that breaks continuity and stability. By embedding filters into $\mathbb{RP}^N$, we obtain the minimal geometric extension that resolves this discontinuity. This embedding ensures mathematically defined scale invariance, satisfying $embed(cF) = embed(F)$ for $c>0$, which cannot be guaranteed by simple L2 normalization.
>
> 2. A geometrically principled view of gradient-based importance.
>
> Projective offsetting enables pruning to be interpreted as movement within projective space, enabling filter importance to be defined by the angular displacement induced by a one-step gradient update. PROscore measures this geometric movement directly and cannot be derived from existing heuristic direction-based scores, which lack a normalization-free and scale-invariant geometric formulation.
>
> 3. Stable importance estimation across diverse scenarios, supported by experiments.
>
> The projective embedding makes importance evaluation stable across a wide variety of scenarios (e.g. Figure 1). Empirically, this geometric foundation leads to strong and consistent performance: on DeiT, IPPRO clearly outperforms SNP (ECCV 2024), and it also demonstrates robust performance on modern large-scale models such as LLaMA.

---

> > ### Author Response · Authors · 2025-11-24
> >
> > >  A) When does it stop taking the filters out from the net?
> > >
> > In importance-score–based pruning, including LLM-Pruner and most prior work, pruning simply stops when the target pruning ratio is reached. This is a standard convention in the literature, and IPPRO follows exactly the same setting.
> >
> > > B) What happens if all filters are removed from a layer and how does the approach handle layer collapse?
> > >
> > Layer collapse only occurs under global pruning. We briefly mentioned this in Appendix C.3 and will clarify it more explicitly.
> >
> > > C) Does it need pre-trained model always?
> > >
> > Yes, this is standard for structured pruning. Pruning without a pretrained model is effectively “training a smaller model from scratch,” which is not considered pruning. IPPRO follows the same widely adopted practice.
> >
> > > Projective geometry seems the key idea of the paper. Yet, there is hardly any clarity in the paper why and how does it help. The description seems too superficial and cursory.
> > >
> > you can see the summary
> >
> > > Algorithm 1 does not add any value, in my opinion. Rather the authors should use the space to better justify why and how of projective geometry.
> > >
> >
> > Algorithm 1 clarifies how projective embedding is performed and how gradients are collected.
> > It is intended to prevent implementation ambiguity and is therefore a meaningful part of the method description.
> >
> > > Results: Finally, the results are nowhere close to the state of the art. For example, on CIFAR 10, CURL (Neural network pruning with residual-connections and limited-data, 2020), SPvR (SPvR: Structured Pruning via Ranking , 2025), Hrank ( Hrank:Filter pruning using high-rank feature map, 2020), OTOv2 achieves the same performance as the proposed method with 40% (absolute) lesser parameters. Overall, I think the proposed approach uses much more parameters than the state of the art models to achieve the same performance.
> > >
> >
> > Our method shows strong performance across modern architectures (DeiT, LLaMA-7B/2-7B) and segmentation models (DeepLabV3), outperforming specialized methods such as SIRFP despite IPPRO being architecture-agnostic.
> >
> > Regarding CIFAR-10, our table reports parameter and FLOPs reductions. For example, HRank achieves 93.5% accuracy with a 29% reduction in FLOPs (17% reduction in parameters), whereas IPPRO achieves 94% accuracy with a 49% reduction in FLOPs (49% reduction in parameters). Based on these numbers, it is unclear how our performance was judged as “similar” to these methods.
> >
> > Additionally, among the methods the mentioned (CURL (2020), SPvR (2025), HRank (2020), and OTOv2) the only overlapping experimental setting with our paper is ResNet-50 on ImageNet-1k. Therefore, we would like to ask on what basis the reviewer concluded that our method underperforms these approaches. To clarify this point, we briefly compared our method with SPvR, and the results are summarized in the table we attached below.
> >
> > |Method ResNet50-ImageNet1k|baseline acc|pruned acc| accuracy diff | pruned params (%)|
> > |--|--|--|--|--|
> > |Hrank|76.15 | 74.98 | -1.17 | 36.6% |
> > |SPvR| 76.32 | 75.58 | -0.74 | 40% |
> > |IPPRO| 76.15 | 76.21 | +0.06 | 46.4%|
> >
> > From this comparison, SPvR reports 75.58% accuracy with 40% parameter reduction, whereas IPPRO achieves 76.21% accuracy under 46.4% parameter reduction. In other words, IPPRO performs better both in terms of parameter reduction and accuracy. Furthermore, the official pretrained ResNet-50 model on ImageNet-1k has a top-1 accuracy of 76.15%, whereas the SPvR paper reports using a pretrained checkpoint with 76.32% accuracy. Since this pretrained weight does not match the official baseline, we do not consider this a fair comparison. We do not consider this to be a fair or verifiable comparison, which is why we chose not to include such results in the main paper.
> >
> > As mentioned earlier, the CNN pruning benchmark currently faces a consistency issue:
> > CIFAR models lack standardized pretrained checkpoints, and many ResNet-50 pruning papers apply additional “good tricks” (e.g., stronger augmentation, modified fine-tuning, altered training pipelines) that deviate substantially from the official training setups. Such discrepancies prevent meaningful or fair comparisons—unlike modern architectures such as ViT or LLaMA, where standardized training and fine-tuning protocols are strictly maintained.
> >
> > For fair comparison, we intentionally followed the same fine-tuning recipe used for the baseline pretrained models. In contrast, many CIFAR/ResNet pruning papers modify the original training setup, which means their reported numbers are not directly comparable. For this reason, we placed such results in the appendix rather than the main text.
> >
> > If you still believes those modified settings still represent a fair comparison, we are fully willing to run additional experiments under the same altered recipes and update the results accordingly.

---

> ### Comment · Reviewer_4m1G · 2025-11-28
> **Overall response**
>
> I keep my score unchanged.

---

### Meta-Review · Area_Chair_DPSS · 2026-01-05

**Summary:**

This paper proposes IPPRO, a structured pruning method based on projective space. Although the authors provided additional explanations and experimental results in their response, the core concerns raised by the reviewers remain unresolved. The main issues are as follows:

1. Insufficient Elaboration of Methodology and Theoretical Contribution
Reviewers 4m1G and z984 both pointed out that the description of the core method is unclear and difficult to reproduce. Although the authors supplemented their explanations in the response, the theoretical justification for why and how projective geometry improves pruning remains weak, failing to adequately demonstrate its fundamental advantage over existing direction-based or normalization-based methods.

2. Doubts Regarding Experimental Comparisons and Performance Claims
Reviewers 4m1G and 8L8K questioned whether the performance reaches SOTA, noting that improvements on some benchmarks are marginal. While the authors argued that comparison baselines were unfair, they did not provide overwhelming evidence of clear superiority over the latest methods under their claimed fair settings. Comparisons with recent 2025 works like VBP are also insufficient.

3. Concerns About Manuscript Quality and Review Trust
The authors questioned the professionalism of Reviewer 4m1G's review. While this is a matter of appeal, it does not alter the substantive weaknesses identified by that reviewer regarding unclear methodological description and failure to achieve SOTA results. Additionally, multiple reviewers noted issues such as grammatical errors and unclear presentation.

While the paper attempts to introduce projective geometry to improve pruning, it fails to provide a sufficiently convincing contribution in any single dimension: theoretical depth, performance superiority, or exposition clarity. Given the unresolved fundamental issues outlined above, the decision is to reject the submission.

**Reviewer Concerns:**

The authors' response clarified methodological details (stopping criteria, need for pre-training), supplemented latency data and comparisons with VBP, and explained mathematical notation and the parameter-injection mechanism.

Unresolved Core Issues
1) Insufficient Theoretical Innovation: Multiple reviewers considered the application of projective geometry superficial and failed to demonstrate its fundamental theoretical advantage over existing direction-based or geometric pruning methods.
2) Unclear Method Description: The original description was still criticized as difficult to understand and reproduce; the response did not fundamentally improve the clarity and self-consistency of the exposition.
3)  Limited Empirical Improvement: Performance advantages on key benchmarks were marginal , and no decisive SOTA-leading advantage was demonstrated under standardized, fair settings.

While the rebuttal addressed some technical questions, it failed to respond to fundamental criticisms regarding the substantiveness of the theoretical contribution, clarity of the method, and significance of the empirical results.

**Reviewer Scores:**

4m1G (Initial Score: 0 )
Projected Final Score: 0
This reviewer had fundamental criticisms regarding the paper's clarity, method description, and results. The author's response failed to alleviate the core concerns (unclear method explanation, results not meeting SOTA), and their stance is expected to be further reinforced during discussion.

z984 (Initial Score: 4)
Projected Final Score: 3-4
This reviewer questioned the theoretical novelty of the method, viewing the projective geometry as over-engineering. The author's response did not effectively counter this core criticism. After observing other reviewers' persistent doubts about the theoretical foundation and experimental performance, the reviewer's score is not expected to increase and may remain at or drop to a clear reject level.

VazZ (Initial Score: 6)
Projected Final Score: 5-6
This reviewer acknowledged the paper's framework and experiments but also pointed out limited theoretical gains and marginal performance improvements. While the author defended these points, the strong criticism from other reviewers regarding the same strengths may shake their confidence. The score may fluctuate slightly around the threshold, but a significant increase is unlikely.

8L8K (Initial Score: 2)
Projected Final Score: 2-3
This reviewer questioned the theoretical justification and noted limited performance gains. Although the author supplemented latency data and theoretical references, they ultimately acknowledged that the core contribution does not lie in massive performance advantages. This may slightly improve understanding but is insufficient to change the initial reject judgment based on "incremental contribution."


The discussion highlighted deep unresolved disagreements regarding the paper's theoretical core contribution and empirical significance. Reviewers with negative scores had their positions reinforced, borderline reviewers found no reason to raise their scores, and the only moderately positive reviewer's confidence may have been undermined. It is projected that the discussion would strengthen the consensus for rejection.

---

### Decision · Program_Chairs · 2026-01-26

Reject